# The Presence of Micro- and Nanoplastics in Food and the Estimation of the Amount Consumed Depending on Dietary Patterns

**DOI:** 10.3390/molecules30183666

**Published:** 2025-09-09

**Authors:** Aleksandra Duda, Katarzyna Petka

**Affiliations:** 1Department of Fermentation Technology and Microbiology, Faculty of Food Technology, University of Agriculture in Krakow, ul. Balicka 122, 30-149 Krakow, Poland; 2Department of Plant Products Technology and Nutrition Hygiene, Faculty of Food Technology, University of Agriculture in Krakow, ul. Balicka 122, 30-149 Krakow, Poland; katarzyna.petka@urk.edu.pl

**Keywords:** microplastic and nanoplastic intake, food contamination, Europe, nutrition models, lacto-ovo-vegetarian diet, western diet, Mediterranean diet, food pyramid

## Abstract

Micro- and nanoplastics (MNPs) are becoming an increasingly common environmental pollutant. They have been detected in fruit, vegetables, drinking water, seafood, meat, dairy products, and cereals, with particularly high levels often being found in processed foods. The presence of MNPs varies significantly depending on the type of food, geographical region, method of food preparation, and packaging materials used. Of the three main routes of human exposure to MNPs, ingestion is the most important. This article provides a comprehensive review of food contamination by MNPs, including an assessment of the impact of various factors on the MNP abundance. For the first time, it also evaluates the differences in MNP intake among individuals following three typical European dietary patterns: the Mediterranean, Western, and lacto-ovo-vegetarian. The lacto-ovo-vegetarian diet was found to result in the highest MNP intake (69.1 × 10^6^ particles/day), almost doubling that of the other tested patterns. This is mainly due to the very high proportion of fruit, vegetables, legumes, and nuts in daily meals. Taking into account both health concerns and MNP quantity consumed with meals (37.5 × 10^6^ particles/day), the Mediterranean diet is the healthiest. The review also highlights the need to raise awareness of food-related sources of MNPs.

## 1. Introduction

For many years, plastics have successfully displaced other materials, such as glass, paper, metals, cotton, wood, and even concrete, due to their durability, lower cost of production and transport, lower thermal conductivity, better barrier properties, and the unlimited range of shapes and colours they can be moulded into [1]. Plastics are used to manufacture various types of packaging and construction materials, components, or entire devices, machines, furniture, and equipment for homes, offices, shops, public facilities, and everyday items. They are also used to make sports equipment, toys, road surfaces, fertilisers, marine and ocean fishing equipment, clothing, disposable medical equipment, etc. [2,3]. However, the slow degradation or abrasion of plastic items, as well as the improper management of plastic waste, leads to it ending up in the natural environment, where it can partially degrade into hazardous substances. This is why the scientific community has recently been inundated, on a previously unprecedented scale, with publications demonstrating the presence of plastic degradation by-products in the human environment.

The term ‘microplastic’ was first coined in 2004 by Thompson et al. [4] to describe microscopic fragments of plastic debris (~20 µm in diameter), resulting from the degradation of plastic items. In recent years, awareness of the negative impact of microplastics (MPs; microplastic particles = plastic particles < 5 mm) and nanoplastics (NPs; nanoplastic particles = plastic particles < 1 μm) on living organisms, including humans, has grown [5,6,7], as has knowledge of methods for their analysis, detection, and removal [8]. Micro- and nanoplastic particles (MNPs) are present in the environment in the form of particles of various sizes and shapes. The most common types found are microfragments (irregularly shaped), films (flakes, flat and thin), fibres (filaments, long and thin), polymer chains, foams (sponge-like), and pellets (spherical granules and beads) [9,10,11,12,13,14]. The most important polymers used to manufacture various plastic items which can enter the environment (e.g., soil, air, water) and living organisms in the form of MNPs include several dozen compounds, such as: polyurethane (PU), poly(vinyl chloride) (PVC), polyethylene (PE), polypropylene (PP), polystyrene (PS), polyethylene terephthalate (PET), polyamide (PA), polycarbonate (PC), polyester (PES), poly(methyl methacrylate) (PMMA), ethylene-vinyl acetate (EVA), high-density polyethylene (HDPE), low-density polyethylene (LDPE), polyvinyl alcohol (PVA/PVOH), polysulfone (PSF/PSU), acrylonitrile butadiene styrene (ABS), polyacrylonitrile (PAN), cellulose acetate (CA), polytetrafluoroethylene (PTFE), polyoxymethylene (POM), styrene-ethylene-butylene-styrene (SEBS), expanded polystyrene (EPS), thermoplastic elastomer (TPE), poly(fumaronitrile:styrene) (FNS), chlorinated polyethylene (CPE), as well as Nylon 6 (polyamide 6), Nylon 66 (polyamide 66), epoxy resins, vinyl esters, silicones, and polyethersulfon [15].

MNPs are ubiquitous environmental pollutants, found in products related to almost every sphere of life. Consequently, their concentration in the environment is constantly increasing [16]. Many studies report the presence of MNPs in various types of water, in particular in seawater [17,18,19,20], air [21,22] and soil ecosystems [6,23,24,25,26], from where they can easily penetrate plant and animal tissues and then enter into the food chain [11,27,28,29,30,31,32] or the human body [33,34,35]. Numerous laboratories are attempting to estimate how many MNPs enter the human body. However, the final figures depend on various factors, including the geographical location of the people being studied and the level of environmental, water, and food contamination in their surroundings, as well as their age, size, cultural and religious heritage, eating and hygiene habits, lifestyle, wealth, and how frequently they engage in activities that release MNPs into the environment. Many studies have found that environmental exposure to MNPs can be harmful to living organisms, including humans. The harmful or even toxic effects of MNPs on the morphology, physiology, biochemistry, and genetics of plants and animals [5,7,36,37,38,39,40,41,42,43,44,45,46,47] have been demonstrated. Human health and physiology are also affected by MNPs, including reproduction, endocrine and immune system function, and coexisting microbiota [48], as was described in more detail in a later section of the review (Section 7).

According to analyses conducted in 2019 [49], the amount of plastic consumed by one person per year ranged from 39,000 to 52,000 particles of MNPs, with this figure continuing to rise. In 2021, another team of scientists estimated the global average rate of microplastic ingestion (GARMI) [50]. Through complex calculations and simulations and the use of three consumption scenarios, it was shown that, on average, one person could potentially ingest 0.1–5 g of MNPs per week worldwide. To give a better idea, this can be compared to consuming the same amount of plastic as is found in a plastic bank card. However, it should be emphasised that this result may differ significantly from the actual plastic consumption of people in different countries and cultures, as several assumptions were made when calculating the GARMI. Among other things, for the calculations of the average mass of microplastic particles, it was assumed that all ingested plastic particles were of uniform shape and size less than 1 mm. In addition, to calculate the minimum and maximum number of MNPs ingested by one person, the average values from publications that assessed MNP contamination in only a few selected products (water, shellfish, fish, salt, beer, honey, and sugar) were taken into account. Some of these products are not typical components of everyone’s diet. Others, which are an important component of many diets (such as vegetables and fruit), were omitted.

Considering all of the above, as well as the differences in MNP content in food products and the differentiated consumption of various products depending on the country and consumer wealth, it is clear that the amount of plastic consumed by a given person will vary greatly and depend on many factors, including diet. Therefore, the aim of this review is to provide a completely new perspective on the problem of the presence of MNPs in food products, using the food pyramid and the differences between various European dietary patterns as a basis. To this end, a comprehensive literature review was conducted on the presence of MNPs in various food products typical of different European countries. Three dietary patterns were selected for use in assessing MNP intake with various diets. Attention was also paid to the sources of MNPs and activities that may increase or reduce human exposure to them.

## 2. Sources of MNPs in the Environment and Pathways of Contamination

Plastic particles are present throughout the globe in all environments, regardless of latitude, climate, or human presence, posing a significant risk of exposure to all living organisms, including aquatic and terrestrial organisms, such as microorganisms, plants, animals, and humans [51]. They mainly enter the environment due to the improper plastic products’ end-of-life management (Figure 1) [52].

Plastic particles have been found in water, air, and soil. When the review mentions the presence of MNPs in water, this refers to any form of water and any type of reservoir or aquatic environment. This includes all the world’s seas and oceans [53,54,55,56,57,58], the Mariana Trench [59], the Great Barrier Reef Marine Park [60], groundwater [61], inland waters such as the Black Sea [62], small and large rivers [63,64,65,66,67,68,69], mountain rivers [70], lakes [71,72], as well as snow and stream water in the Arctic [73], Antarctica [58,74,75], Mount Everest [76], and alpine glaciers [77]. MNPs enter water reservoirs from the air during rainfall or monsoons [57,78]; plastics used in aquaculture [79,80] and for fishing [81,82]; clothing made from new or recycled plastics and microfibres (particularly synthetic fleeces) during their household washing and drying [83,84,85,86]; cosmetics and personal hygiene products [87,88]; and as a result of soil erosion [89]. The amount of particles released from clothes during washing and drying depends on the type and texture of the fabric, as well as the washing and drying parameters (time, temperature, and whether the machine is top- or front-loaded) [83,84,86].

Plastic particles are present in the air, both indoors and outdoors, with higher levels of MNP contamination found in indoor air [16,90,91]. MNPs have been detected in the atmosphere above densely populated cities and heavily industrialised areas [21,92,93], as well as over the oceans [94,95] and in the Amazon basin [96]. They mainly enter the air from worn or mechanically damaged plastic surfaces [97,98,99], but they can also be carried by ocean waves or soil particles [20,98,100,101]. Both cultivated and uncultivated soil can contain plastic particles, regardless of its fertility, agrotechnical processes, or fertilisation; however, pollution levels may vary seasonally, depending on the amount and frequency of rainfall, wind strength, and farming activities [23,26,102,103,104,105]. MNPs have also been detected in marine, lake, and river sediments [63,106,107], on beaches [108,109,110,111], and even in desert sand [112,113].

The presence of plastic in the environment is possible because of its five main characteristics and properties: it is a synthetic material with a high polymer content, it is a solid particle, it measures less than 5 mm, it is insoluble in water, and it does not undergo complete degradation. Sources of MNPs can be classified as primary or secondary [114,115].

**Figure 1 molecules-30-03666-f001:**
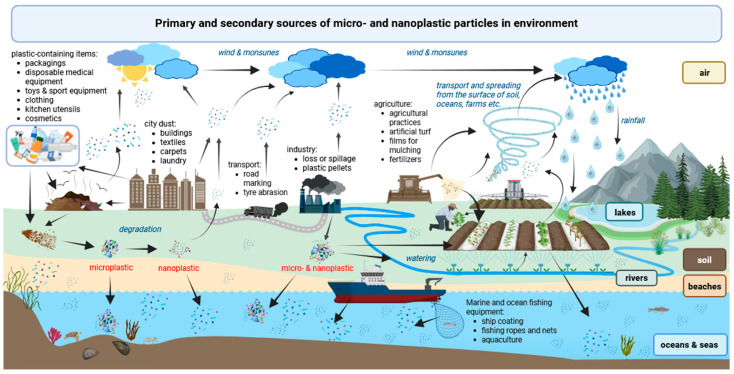
Sources and routes of contamination of the environment by micro- and nanoplastics. Adapted from [20,23,26,78,94,101,102,114,116,117,118]. Graphic prepared in BioRender [119].

Primary sources include the intentional industrial use of various types of plastic, which can enter the environment directly in the form of small particles, e.g., as a result of abrasion, weathering, or unintentional loss or spillage during the production, transport, use, maintenance, or recycling of products containing plastic or plastic pellets (mainly unintentionally) [114]. They also include plastic microbeads that are intentionally added as abrasives to toothpaste, exfoliating cosmetics, and toiletries (e.g., shower gels) [35,87,88,120], from which they are washed out during normal product use. The majority of unintentional plastic starts its life as pellets used later for the production of plastic items (85%), synthetic textiles (12%), or synthetic rubber in tyres (2%) [114]. Secondary sources are formed unintentionally, usually as a result of larger plastic items degrading under the influence of environmental factors (sunlight, temperature, and salt water), mechanical forces, normal use (e.g., cleaning, wear and tear), or the activity of microorganisms. This fragmentation produces particles measuring less than 5 mm in diameter (MPs) or smaller than 1 μm (NPs).

The following sources have been identified as the main contributors to the release of MNPs into the environment (Figure 1):Mechanical fragmentation of plastic products during normal, daily use or as a result of weathering, abrasion, and pouring of particular objects, especially when improperly recycled (synthetic shoe soles, toys, synthetic kitchen utensils, medical devices, infrastructure components, buildings, equipment, carpets, furniture) [99,121,122,123,124],Vehicle tyres due to abrasion while driving [97,116,125],Broken plastic fishing ropes and nets [126],Synthetic textiles—through abrasion and shedding of fibres during laundry washing and drying [127,128],Artificial turf [129], plastic film used as mulch [130,131], and plastic-coated fertilisers [103,132,133] used in agriculture,Cosmetic cleansers and other personal care products [87,88,120]—by direct introduction of the plastic particles into wastewater streams from households, hotels, hospitals, and sports facilities, including beaches,Paints used for road marking (e.g., paint, thermoplastic, preformed polymer tape, and epoxy degraded due to weathering and abrasion by vehicles) [134,135] and marine coatings (due to weathering and incidents during application, maintenance, and disposal) [114].

It should be highlighted that plastic items and their fragments, as well as microplastics and nanoplastics derived from them, accumulate in the environment primarily due to their high resistance to biodegradation. The time it takes for plastics to degrade depends on the polymer that makes up the plastic particles, the environment [136], and the factors at work, such as UV rays, temperature, oxygen, salt water, soil, and—to a lesser extent—the presence of microorganisms. The subsequent stages of decomposition can last from several months to many years [52,136]. Degradation processes can be abiotic, caused by mechanical, physical, or chemical factors, or biotic, dependent on living organisms [52,137,138]. Among the most important are as follows:Photodegradation (photocatalytic degradation, photooxidation)—under the influence of light (both visible and UV, usually from sunlight) in outdoor exposure conditions [139,140,141,142];Physical degradation—caused by mechanical action, for example, during cleaning, transport, laundry, drying, or by shear stress forces [83,120,143,144,145];Thermal degradation—under the influence of high temperatures, accelerated by the presence of oxygen (thermo-oxidative degradation) [14,121,146,147,148];Hydrolytic degradation—occurring in an aqueous environment, when ester or amide bonds undergo cleavage [14,149], but which can also be accelerated by acids [150],Biodegradation and biodisintegration—the action of living organisms (usually microbes) resulting from the activity of their enzymes, which cause hydrolysis and chain cleavage, as well as fragmentation of larger particles into smaller ones [52,151].

## 3. Entry Routes of MNPs into the Human Body

It is recognised that MNPs can enter the human body via three key routes. These are inhalation, dermal penetration, and ingestion [115].

### 3.1. Inhalation of Airborne Plastic Particles

An important role is played here by both indoor breathing, where the air is polluted with particles originating, among other things, from clothing [152], face masks [153], carpets and furniture [90,154], as well as outdoor air, where MNPs particles come from tyre abrasion [97], building construction materials [16], aerosols formed by ocean waves [20,155,156], particles from fertilisers [104,105,132], sewage sludge [157,158], wastewater [159], atmospheric fallout [118,160], and synthetic textiles during their production [161], wearing, or industrial washing and drying of fabrics [76,162,163]. Numerous studies have indicated that indoor air is more polluted with plastic particles than outdoor air [92,164].

Dris et al. [90] assessed MP concentration in the outdoor air at 0.3–1.5 particles/m^3^, while indoor concentrations were 0.4–56.5 particles/m^3^. Therefore, assuming that one person inhales ~6 dm^3^ of air per minute, humans could be exposed to over 48000 microparticles each day, with PES (81%), PE (5%), and nylon (3%) being the dominant indoor polymers [164]. It is obvious that the smaller the particles, the deeper they penetrate the respiratory system [165]. The particles measuring 0.1–10 μm are particularly harmful, as they reach the pulmonary alveoli and may migrate to the circulatory or lymphatic system. These particles can also cross the blood–brain and placental barriers [166,167]. Studies of human lung tissue obtained during autopsy have reported the presence of PE and PP particles smaller than 5.5 μm, as well as fibres ranging in size from 8.12 to 16.8 μm, in 13 out of 20 tissue samples [168].

### 3.2. Dermal Penetration of Nanoplastic Particles

Microplastic particles, and to an even greater extent NPs, can penetrate the skin during bathing and daily personal hygiene, either from water containing MNPs or from personal hygiene products and cosmetics, face masks, or dressings [88,169,170,171], as well as from daily consumables such as erasers and pen accessories [172]. The stratum corneum forms a physical protective barrier for the skin, and—due to the hydrophobic properties of plastic particles—the absorption of NPs through human skin in water is rather limited. It is assumed that only NPs smaller than 100 nm can penetrate the stratum corneum and enter the dermal or even subcutaneous layers of the skin [33,167,173]. Other possible entry routes include hair follicles, sweat gland outlets, intercellular pathways, or via injured skin areas [174,175,176,177]. Penetration of nanoparticles was also observed through barrier-impaired skin, e.g., when some components of cosmetics were present [170]. Oleic acid, ethanol, and oleic acid–ethanol can enhance the transdermal delivery of zinc oxide nanoparticles, as they increase the intercellular lipid fluidity or extract lipids from the stratum corneum [178].

### 3.3. Ingestion of MNPs

Micro- and nanoplastic particles, independently of their size, can enter the human gastrointestinal tract and the body by direct ingestion or via trophic transfer.

Direct ingestion mainly occurs when humans consume potable water containing MNPs, as well as various soft drinks, alcoholic and non-alcoholic beverages, or meals prepared using contaminated tap water. MNPs can also be swallowed accidentally from non-food sources during oral hygiene and tooth brushing from sources such as toothpaste, toothbrushes, orthodontic implants, and denture materials [88,179,180,181,182]. There are a lot of studies reporting the presence of MNPs in treated and untreated tap water [18,19], groundwater [61,183,184], bottled mineral water [17,82,185], beer [186], wine [187], soft drinks [188,189,190], as well as in water from distributors, public fountains, water kiosks, and various refilled packaging [184,191,192]. MNPs can be released into bottled beverages from bottle walls, PET bottle necks, and HDPE caps [144,193], into wine from PE stoppers [194], into coffee and tea from plastic teabags or capsules [149,195,196], from disposable cups [195,197], or they can be washed out when boiling water in a kettle [198]. In addition, MNPs may originate from sources of drinkable water (e.g., rivers, lakes, or groundwater) that are contaminated with plastic particles [65,199]. These particles may persist in tap water even after treatment processes [200,201,202,203] or enter the water during purification processes [204]. Weisser et al. [205] reported that MNPs can be detected in bottled mineral water because particles present in groundwater can contaminate bottles when they are cleaned and filled for reuse.

Trophic transfer of MNPs occurs when aquatic organisms or food prepared from plant or animal raw materials that have ‘absorbed’ MNPs from the environment (water, air, soil) or from feed are consumed. The accumulation of MNPs in the tissues of aquatic organisms occurs as a result of the accidental ingestion of micro- or nanoplastic particles from polluted water reservoirs (Figure 2). Plankton and other filter-feeding organisms (such as oysters, shrimp, and crabs) are food for predators (such as fish and mammals), which makes them the first stage in the entry of microplastics into the food chain [27,29,67,206,207,208,209,210,211,212]. MNPs can also penetrate plant tissues, entering them either from the soil via uptake by plant roots or through uptake by leaves from irrigation water or polluted air [36]. These particles then enter the tissues of herbivorous animals, which are subsequently used as raw materials in food production and are ultimately consumed by humans [28,31,213,214,215]. Additionally, plastic particles can enter the food chain through food additives, such as salt and sugar [216], as well as through the use of plastic packaging, tableware, and kitchen accessories [121,123,217] used for food preparation and meal serving.

Of the three routes of human exposure to MNPs mentioned above, both inhalation and skin penetration depend on the size of the plastic particles, whereas ingestion applies to MNPs of all sizes. Moreover, the walls of the human digestive tract have a surface area of approximately 200 m^2^, making it the main route of entry for MNPs into the body. The lungs also have a very large surface area with a very thin tissue barrier that enables nanoplastic particles to penetrate into the capillary blood system and to distribute throughout the human body [115,218]. However, considering the level of indoor and outdoor air contamination with MNPs, their size, and the defence systems of the human body, it can be concluded that inhalation does not play a significant role in human exposure to MNPs [219]. According to Domenech and Marcos [220], the mean inhaled intake of MNPs is about 2.16 × 10^3^ particles per year, and even if these figures are underestimated, the MNP intake by inhalation is still incomparably lower than their intake by ingestion. The same applies to the absorption of MNPs from cosmetics, face masks [153], or clothing through the skin, since the stratum corneum acts as a fairly effective protective barrier [115], and healthy skin that has not been injured is considered an effective barrier to plastic particle penetration. This means that special attention should be given to food and the types of plastic that can contaminate it, as well as the various ways in which food can become contaminated by MNP particles.

The growing awareness of the impact of environmental pollution, including the presence of micro- and nanoplastics in water, soil, air, and food, is reflected in the regulations implemented by the European Union. The infographic below (Figure 3) presents the main legislative acts in the European Union that regulate the monitoring and control of environmental pollution, including micro- and nanoplastics, and establish a methodology to measure microplastic contamination.

## 4. Mechanisms of Food Contamination

The degree of microplastic pollution in foodstuffs depends not only on the raw materials used, the farming techniques employed, and the fodder used to rear livestock, but also on the techniques and apparatus used to process and preserve the obtained raw materials and the materials used in production, particularly those used to package finished foodstuffs [2,14,147]. It also depends on the storage conditions, as well as on the method of food preparation (e.g., heating in plastic packaging [14,147]) and its delivery for consumption (e.g., disposable cups or plates [195], plastic takeaway containers [148,221]).

### 4.1. Kitchen Accessories, Food Preparation, and Food Packaging as a Source of MNPs in Meals

One of the most important pathways by which plastics penetrate food is, of course, through the trophic chain. Micro- and nanoplastic particles from the environment enter and accumulate in the tissues of plants and animals (Figure 2), which are then used as the raw material for preparing human meals. However, it should be emphasised that this is not the only way in which MNPs enter food. Numerous studies indicate the important role of preparing meals at home oneself, including processes involving significant mechanical stress, such as cutting, chopping, grinding, mixing, beating, etc. [121,123,217,222]. The increasing use of plastic kitchen accessories (e.g., cutting boards, spatulas, beaters, spoons, funnels, strainers, whisks, and various bowls) and appliances with plastic elements (e.g., mixers, blenders, and food processors) instead of glass, wood, or metal ones causes these items to be a significant source of MNPs in ready meals [223,224,225,226]. The number of particles released depends on both the material from which these accessories are made and how they are used. Luo et al. [227] reported that 100–300 MNPs per mm per cut along the groove were formed on the chopping board, and approx. 3000 per mm^2^ per cut could be released from a scratched area during food preparation. The amount of MNPs released from the chopping board depended, among other things, on different cutting habits, skills, and strengths. Not only the cutting technique (i.e., knife pressure or impact force on the cutting board), but also the structure of food (including the presence of hard or sharp ingredients such as bones, nuts, seeds, and even salt) can influence the amount of plastic released through abrasion or mechanical damage. The number of particles released during mixing as a result of abrasion of the walls of bowls made of ABS was lower when pure water was mixed than when water containing rock salt was used (591 vs. 1890 particles) [121]. In the study of Habib et al. [228], the authors detected up to 2.47 ± 0.4 mg of MPs per 1 g of fish and 0.39 ± 0.25 mg of MPs per 1 g of chicken samples that had been cut on plastic cutting boards by the vendors in shops. Chickens with bones generated more MPs than chicken fillets, as cutting them required greater force. When cutting boards made of different materials were compared, it was found that PP boards released greater amounts of MPs than those made of PE [229]. The annual exposure of one person to MPs from chopping boards has been estimated at 7.4–50.7 g (corresponding to 14.5–71.9 × 10^6^ particles) for PE boards and approximately 49.5 g (79.4 × 10^6^ particles) for PP boards [229]. On the other hand, PE-based food bags have been proven to release more particles than plastic containers made from PP [230], indicating that the selection of the polymer should be based on the final destination and intended application of the plastic item. Different plastics should be used for water-based foods, and others for fatty foods (oils) or acidic foods. The same applies to the temperature to which plastics are planned to be exposed, as heating in a microwave oven works differently from heating in a traditional oven, and so the temperature reached by plastic items varies [231].

Polytetrafluoroethylene (PTFE) is the trade name for Teflon, which is often used to make non-stick pots and pans. When using these items, the Teflon surface of the cookware may be mechanically damaged, e.g., by scratching with sharp cutlery when stirring food, or it may degrade due to the normal process of thermal ageing [232]. Luo et al. [233] demonstrated the release of over 2 × 10^6^ MNPs from the damaged surface of Teflon-coated pots. It should also be highlighted that PTFE, when overheated (temperature reaching up to 370 °C), can generate significant amounts of perfluoroalkyl carboxylic acids (PFCAs) as a result of PTFE thermolysis [234], and these substances are harmful to human health [235]. Other studies have shown that ceramic-coated cookware releases TiO_2_ and SiO_2_ nanoparticles at concentrations of 10^8^ and 10^7^ particles/dm^2^, respectively, during simulated intensive scrubbing and scratching [236]. Mechanical stress generally promotes the release of plastic particles. The use of a blender for just 30 s has released approx. 0.36–0.78 × 10^9^ MNP particles from its plastic container [237]. A good example of a process involving various factors that contribute to the generation of MNPs is homemade jelly prepared with plastic accessories and utensils. The preparation of jelly involves several stages, including heating, cooling, mixing, cutting, and storage. If plastic accessories were used, particles of PTFE, PE, and PP, as well as fibrous particles ranging in size from 13 to 318 μm, were detected in the final jelly [222]. Assuming that such jelly was consumed daily, this could translate into the ingestion of 2409 to 4964 microplastic particles per year. Importantly, the use of kitchen utensils that were not plastic did not result in MPs being introduced into the jelly.

Analysis of kitchen bowls made from different types of plastic and used with hand mixers has shown that various amounts of microplastic particles are released through abrasion when mechanical or thermal stress is applied, depending on the material used for production (melamine, ABS, PP, LDPE, PS, and styrene-acrylonitrile (SAN)) [121]. Containers made from melamine released the highest number of MPs (898), while the lowest number of abraded particles were reported for an LPDE bowl. Thermal degradation of ABS polymer conducted at 200 or 250 °C led to the release of various hazardous substances, such as styrene, acrolein, benzaldehyde, methylstyrene, and acetophenone. Acetaldehyde, as well as acetic and formic acids, was released from LDPE, while PS thermal degradation mainly released benzaldehydes and styrene. Melamine polymer resulted in the release of formaldehyde and methanol, and PP produced acetone, as well as formic and acetic acids, particularly at 250 °C. All of the aforementioned substances have been proven to exert harmful, irritating, toxic, or even carcinogenic impact on human cells and tissues when ingested, inhaled, or touched [238]. By comparing particle amounts and the hazard classifications of the main components, LDPE seems to be the safest plastic for use in a household mixing bowl [121]. Moreover, considering the release of formaldehyde and melamine monomers from various melamine tableware [239], some producers recommend not using the melamine-containing kitchen accessories in microwaves and avoiding heating them above 71 °C (170 °F).

The release of MNPs from packaging into food is affected by the type of heat treatment used, as well as by large temperature changes. Storing packaged food at low temperatures (typically −21 °C in a freezer, 4 °C in a refrigerator) and then reheating it (approx. 40–50 °C) or exposing it to high temperatures of up to 250 °C during thermal food processing (e.g., boiling, baking, stewing, frying, reheating in a microwave oven, pasteurisation) may affect the integrity of the plastic used for the packaging and dishes involved in these activities. In a study by Hussain et al. [230], the release of MNPs from plastic containers and reusable food bags was assessed when they were heated in various ways in contact with deionised water and 3% acetic acid (to simulate water-based and acidic foods). The authors showed that heating in a microwave oven resulted in the release of more MPs and NPs into food than storage at room temperature or in a refrigerator. Just three minutes of heating in a microwave oven led to the release of as many as 4.22 × 10^6^ MPs and 2.11 × 10^9^ NPs from a 1 cm^2^ plastic surface. Similar quantities of MNPs were also released during storage in a refrigerator or at room temperature, but only if the samples were stored for more than six months. The quantity of MPs released depends on the food properties, including its acidity and lipid content. Liu et al. [240] demonstrated that the greatest abundance of MPs was present when acidic high-oil simulants were used for PP samples (1311.33 ± 262.22 items/piece) and for PE samples (1414.00 ± 214.52 items/piece). The abundance of MPs released from plastic samples by different food simulants was found to follow this order: acidic high-oil > alkaline high-oil ≈ acidic low oil > acidic water ≈ alkaline water ≈ high salt > neutral water. This confirms that low-pH and high-oil content promote the release of MPs.

Shi et al. [198] demonstrated that new plastic kettles may release up to 3.5 × 10^7^ microplastic particles per 1 dm^3^ of hard water during boiling, which is consistent with the results of Sturm et al. [241]. Just pouring boiling water into a plastic container can result in the release of large amounts of MNPs. After 60 min of mixing, plastic packaging, cups, transparent boxes, and expandable boxes for food that had been soaked in hot water (100 °C) released 1.07 ± 0.507, 1.44 ± 0.147, 2.24 ± 0.719, and 1.57 ± 0.599 million microparticles and submicron particles per 1 cm^3^, respectively, with the submicron fraction being dominant [242]. Furthermore, organic substances and heavy metals (mainly arsenic, chromium, and lead) were detected in the leachates from the plastic packaging, cups, and expandable boxes, with maximum concentrations of 2.1 ± 0.85 mg C/dm^3^ and 4.2 ± 0.32 ng/dm^3^, respectively. This indicates a potential risk associated with these materials when used to store hot food or beverages. Packaging composed of a PS plate and wrapped film caused higher amounts of microplastic particles to be released into fish when stored for one week at −20 °C. Meanwhile, fish packed with a chitosan film and a PS plate and wrapped film had the lowest level of MPs at both 4 °C and −20 °C, over the entire three-week storage period [243]. Zangmeister et al. [244] found that single-use food-grade nylon bags and hot beverage cups lined with LDPE released up to 10^12^ nanoparticles per 1 dm^3^ when exposed to hot water. The number of particles released depended on the initial water temperature, with food-grade nylon releasing seven times more particles than single-use beverage cups.

According to Zhou’s study [245], people may unconsciously ingest 37,613–89,294 microplastic particles each year by using one plastic cup every 4–5 days. Of course, not everyone drinks coffee or tea from a plastic takeaway cup, but even when tea is brewed at home using a plastic teabag, a huge number of micro- and nanoparticles are released into the drink. The teabags were analysed in studies by Hernandez et al. [149], who estimated that steeping a single empty teabag made from nylon or PET at 95 °C for 5 min releases approximately 1.16 × 10^10^ MPs and 3.1 × 10^9^ NPs into a single cup of tea. The authors estimated that 16 μg of micron-sized particles (1–150 μm) and submicron plastic particles (<1 μm) can be ingested with one cup of tea prepared with one teabag. Analysis of microplastic release from coffee bags made of various materials (PE, PP, PET, and artificial silk) showed that a single plastic coffee pod steeped in water at 95 °C for five minutes can release over 10^4^ microplastic particles per cup, over 80% of which are Rayon. Assuming we drink three to four cups of coffee a day, each person consumes approximately 5 × 10^4^ particles from coffee alone a day [196]. The material used to produce teabags strongly affects the amount of MNPs released, with biodegradable bags typically releasing smaller quantities [246]. However, completely opposite results were obtained in Yang’s study [247]. Biodegradable, single-use paper cups (SUPCs) made of polylactic acid (PLA) released 4.2 times more particles in total, and 3.6 times more microplastic particles than conventional PE single-use paper cups (PE-SUPCs). Furthermore, substantial levels of cellulose microfibres (CMFs) were identified in tests involving PLA-SUPCs, but not in those involving PE-SUPCs. Banaei et al. [248] proved that PLA-based teabags can release approximately 1 × 10^6^ nanoparticles per teabag.

It should be mentioned that the results from different studies may vary due to the various methods used to calculate the amount of particles. The most commonly used methods include scanning electron microscopy (SEM), Raman imaging, Fourier transform infrared spectroscopy (FTIR), stereomicroscopy or fluorescence microscopy, nanoparticle tracking analysis (NTA), and atomic force microscopy (AFM) [249].

A particularly important role in the release of plastics into food is also played by items used for transporting and distributing food, as well as for serving food in restaurants and catering services. These items include food containers, trays, and packaging films made from LDPE, PP, EPS, or extruded polystyrene (XPS), which are used to distribute meat, cheese, salads, fruit, and ready-to-eat food. Other items include packaging for takeaway food and beverages containing PP, PS, PE, PET, or EPS; snack and sweet wrappers made from PP; PET bottles for water; cartons for soft drinks and fruit juices; and HDPE bottles for milk [11,13,151,225,250]. Du et al. [221] examined take-out food containers made from different types of plastics (PP, PS, PE, and PET) and found that they all contained microplastic particles originating from atmospheric fallout or the inner surface of the containers (in the form of flakes). The polymers identified matched those used to make the containers, but also included PES, Rayon (RY), acrylic, and nylon. The highest number of particles was detected in PS containers with rough walls, particularly when subjected to slight mechanical forces. The authors estimated that individuals who order take-out food 4–7 times per week could ingest 12–203 pieces of MPs solely from containers. Even higher values were reported in another study, in which the MPs’ presence in 146 takeaway food samples (including solid food samples such as rice, noodles, bean products, meat, and vegetables, as well as beverage samples such as coffee and bubble tea) was assessed [251]. According to the authors, people who order takeaway food once or twice per week may consume approximately 170–638 plastic particles. The abundance of MPs in take-out food was influenced by food categories (solid > liquid, with rice being the food most contaminated), cooking methods (steaming > frying > roasting > boiling > stir-frying > brewing), and food packaging materials (PS released the highest number of MP particles). These results suggest that the growing popularity of ready-to-eat meals delivered in plastic packaging may lead to increased exposure to MNPs and have negative health effects. No special conditions are required for plastic to be released from packaging. Fadare et al. [252] have demonstrated that new food containers could release MNP particles, most of which were smaller than 50 nm, even when they were simply rinsed with water.

Katsara et al. [253] reported the migration of LDPE and PP from packaging in Cretan Graviera cheese, which was cut into slices, placed in various plastic containers (with or without O_2_ presence), and stored under refrigerated conditions for the next 2–3 weeks. Moreno-Godaliza et al. [254] demonstrated that various commercially available reusable PP containers containing silver can release Ag-containing microplastics when heated in a microwave oven (simulating the daily reheating of meals in reusable containers). The amount of MPs and Ag released varied depending on the type of container, temperature, contact time, acidity of the food, and crystallinity of the container. It is important that in some cases, silver migration to food simulants exceeded the EFSA requirement of 0.05 mg silver/kg of food. Another study proved that the amount of MPs released was affected by the temperature of the water (4 °C, 50 °C, and 80 °C), exposure time (0–20 min), and the material the cups were made of (PP, PS, PE-coated paper cups, and EPS) [255]. The number of MPs ranged from 126 to 1420 particles per 1 dm^3^; the highest counts were observed in PP cups exposed to 50 °C for 20 min. The estimated annual ingestion of MPs from hot and cold beverages in disposable cups was 18,720–73,840. Some plastics, such as EPS or XPS, provided a good thermal barrier and protected against the penetration of oxygen, water vapour, and microorganisms. Consequently, many fresh products (e.g., meat, fish, and cheese) and takeaway meals are distributed on trays made of EPS or XPS. Microplastics originating from XPS (MP-XPS) have been found to enter food at levels ranging from 4.0 to 18.7 MP-XPS per kg of packaged meat [256]. When frozen, glazed, and vacuum-packed icefish placed in PE plastic packages inside a paper case were analysed, more than 90% of samples were found to be contaminated with plastic fragments ranging in size from 50 to 6000 μm, with an average of 0.42 fragments per 1 g of wet sample weight [257].

Many consumers are certainly open to the idea that disposable plastic water bottles can be a source of MPs in the water they drink. However, only a few are aware that the same problem also applies to glass bottles [185] and reusable plastic bottles that we wash and refill [258]. Although it is counterintuitive to think that glass bottles could contain more microplastic particles than plastic bottles, some studies have proven this to be true [259,260]. The amount of MPs released into the water can increase with such a simple action as repeatedly unscrewing and screwing the cap on the bottle [144] or washing it in a dishwasher [261]. The MPs released when opening or closing a PET bottle with a cap made of PP containing PE seals came mainly from the caps, as a significant increase in the number of PP particles (from 100 ± 27 to 185 ± 52 MPs/dm^3^) was reported [262]. Moreover, when PET-bottled beverages intended for outdoor use were studied, sunlight and alkalinity were shown to promote the generation of MPs, mostly of 1–5 μm size [263]. MPs were also released more easily when acidity was combined with temperature or alkalinity with sunlight, with maximum releases of 21,622 ± 2477 particles/dm^3^ and 31,081 ± 7173 particles/dm^3^, respectively. The MPs’ abundance in bottled water from Chinese markets was higher than that detected in tap water (72.32 ± 44.64 vs. 49.67 ± 17.49 items/dm^3^), and microplastics consisted mainly of cellulose and PVC [185]. Interestingly, MPs were present in both PET-bottled water and in glass-bottled water. According to Weisser et al. [205], plastic particles could enter the water during the process of bottle cleaning, filling, and capping, with the latter being responsible for the greatest increase in MP levels. As 81% of MPs resembled the PE-based cap sealing material, abrasion from the seals was identified as the main entry path for MPs into bottled mineral water. Mason et al. [264] analysed one brand of water (Gerolsteiner), packaged in both plastic and glass bottles, and demonstrated that water bottled in glass was less contaminated than the same water packaged in plastic (204 vs. 1410 MPs/dm^3^, respectively). These results indicate that, although some of the microplastic contamination originates from the water source, the packaging itself has a significant impact. Glass bottles are mainly reused on an industrial scale, where they are washed and refilled, while plastic bottles are recycled. During this process, they are processed into pellets, which are then mixed with newly synthesised plastic in various proportions and used to produce new plastic bottles. Therefore, the results obtained in the study conducted by Gambino et al. [260] are not so surprising. The authors compared MPs content in water sold in PET, rPET (recycled PET—a material containing 30% to 50% recycled PET), and glass bottles. On average, water from glass bottles contained significantly more MP particles (8.65 ± 5.39 particles/dm^3^) than water from plastic bottles (5.09 ± 3.28 particles/dm^3^ for PET and 3.31 ± 1.34 particles/dm^3^ for rPET). The microplastic particles differed in size and the polymer composition. Glass bottles mainly released the smallest particles (the size class 20–50 μm constituted 40.49% of all particles), while PET bottles mainly released particles sized 20–50 μm (33.86%) and 50–100 μm (34.92%), and rPET bottles mainly released particles sized 50–100 μm (41.76%) and >100 μm (35.16% of the total). Significantly higher values, although still following the same trend, were obtained in tests of mineral water from Bavaria [17]. The differences may be the result of different analytical techniques. Microplastic contamination was highest in water from glass bottles (6292 ± 10,521 particles/dm^3^), followed by reusable PET bottles (on average 4889 ± 5432 particles/dm^3^) and single-use PET bottles (2649 ± 2857 particles/dm^3^). As expected, the level of microplastic contamination was higher in older reusable bottles (8339 ± 7043 particles/dm^3^) than in new ones (2689 ± 4371 particles/dm^3^). The highest MP level in beverages from glass bottles when compared with plastic bottles and cans was demonstrated also in the newest research of French scientists [259]. As most of the MPs isolated from glass bottles were the same colour as the paint on the outer layer of the metal caps, and as the MPs from both the caps and the bottles were identified as polyester class by FTIR analysis, the authors concluded that painted metal caps are the main source of microplastics in various beverages (cola, tea, lemonade, and beer) sold in glass containers. The differences between particular samples analysed were the result of the brand and water origin, as well as the presence/absence of CO_2_, sweeteners, etc. For example, mineral waters had significantly higher levels of MPs (3.7 ± 1.0 MPs/dm^3^) compared to spring waters (1.6 ± 0.6 MPs/dm^3^), while sparkling water tended to be more contaminated with MPs (3.4 ± 1.0 MPs/dm^3^) than still water (2.4 ± 0.9 MPs/dm^3^), but the latter results were insignificant. Moreover, independently of packaging type, colas with sweeteners had an average content of MPs lower than those without sweeteners (14.3 ± 6.2 MPs/dm^3^ vs. 48.5 ± 30.6 MPs/dm^3^, respectively) [259].

Bottled mineral water usually originates from sources that are protected from pollution, so the better quality of such water is expected. Moreover, water packaged in single-use containers must meet higher microbiological criteria than drinking water available from the tap. However, packaging it in plastic bottles (disposable or reusable) can result in an increased transfer of MNPs to the water, especially when the packaging is handled improperly (e.g., exposure to sunlight, high temperatures, or mechanical stress) or the plastic used for bottle production is thin or of poor quality, or when bottles are washed before filling. Ten brands of still or sparkling mineral water distributed in PET plastic bottles and purchased from different supermarkets in the province of Catania (Italy) were examined, and MP particles were detected in all of the samples [265]. The concentration (mg/dm^3^) of MPs ranged from 100 mg/dm^3^ (still) to 3000 mg/dm^3^ (sparkling), with an average value of 656.8 mg/dm^3^. This translates to an estimated daily intake of 40.1 mg/kg body weight/day for adults and 87.8 mg/kg body weight/day for children. The MP level in bottled mineral water was strongly correlated with its pH. Moreover, hard plastic bottles released larger fragments but in a minor number, whereas the more deformable plastic bottles and a weakly alkaline pH increased the number of smaller MPs. A high level of MP contamination was reported in German bottled mineral water, with values reaching 118 ± 88 particles/dm^3^ in returnable bottles, 14 ± 14 particles/dm^3^ in single-use plastic bottles, and 0 to 253 particles/dm^3^ in glass-bottled water [266].

In light of all the above, kitchen equipment and accessories, as well as containers and films used for serving and storing food, can be considered significant sources of MNPs in food. However, it is worth remembering activities not entirely associated with meal preparation, such as washing and cleaning (washing countertops, appliances, and kitchenware). A dish sponge that is used every day to clean cookware and eating utensils can release particles from both the soft and hard layers, mainly Nylon 6 (PA6) and PET [267]. Even simple, everyday activities such as opening food packaging with scissors, tearing it by hand, cutting it with a knife, or twisting plastic containers, bags, tapes, or caps can generate approximately 0.46–250 MPs/cm^2^. The exact amount depends on factors such as the plastic’s stiffness, thickness, anisotropy, and density, as well as the size of the MPs [268].

Awareness of where MNPs come from in food and during which kitchen operations they are produced in the greatest quantities can help reduce the consumption of MNP particles with food. For human health reasons, it is also extremely important to use materials that are tested and approved for food contact, and to use them in accordance with the manufacturer’s instructions (such as ‘for use in an oven’, ‘suitable for microwave’, or ‘use only at temperatures below 71 °C’).

### 4.2. Plastics Particles Identified in Various Types of Food

As previously mentioned, it has been estimated that the average person unintentionally ingests 39,000–52,000 MNPs per year, which corresponds to consuming up to 5 g of plastic per week [49,50]. However, this average value is the result of a number of factors. These include consumer wealth; access to various raw materials and products; diet and eating habits; food preparation and serving methods; and the level of environmental pollution in the area where the consumer lives.

The types of polymers detected in food also depend on the type of the particles entering the raw materials from water, air, and soil; the production process used and its stages (e.g., filtration, homogenisation, washing, dilution, mixing, filling, capping); and the materials used for food packaging (or take-out containers) as well as kitchen accessories used for meal preparation.

Depending on the thermoplastic resin used, there are six types of plastic materials that are most commonly used to make recyclable plastic packaging. These are polyethylene terephthalate (PET) (known as type 1); high-density polyethylene (HDPE) (type 2); polyvinyl chloride (PVC) (type 3); low-density polyethylene (LDPE) (type 4); polypropylene (PP) (type 5); and polystyrene (PS) (type 6). The last group (type 7) consists of multilayer plastics and other plastics that are not usually collected for recycling [269]. Therefore, it is not surprising that these are among the most commonly detected plastics in food, which are PE, PP, PET, PES, PA, PS, PVC, polyacrylate (PAC), PU, RY, and cellophane (CP) [11,13]. However, the proportions of these polymers vary depending on the food product.

Food guidelines across various nations emphasise the significance of maintaining adequate hydration, advocating that water should constitute the primary fluid component of our diet, with a recommended daily intake of up to 2.5 dm^3^. Water is therefore the most important component of the human daily diet by weight, and is the main source of MNPs in the body. The types of plastic detected in water depend on the type of raw water and the environment surrounding the drinking water intake, including industrial and other human activities. Some researchers suggest that raw water does not necessarily contain fewer MPs than treated drinking water since certain compounds only enter the water during the treatment process. Wang et al. [270] showed that filtration using granular activated carbon enables the removal of up to 60.9% of microplastic particles, mainly of small size, whereas coagulation combined with sedimentation can reduce microplastic particle levels by up to 54.5% (primarily fibres). However, a higher level of polyacrylamide (PAM) was detected in the water after sedimentation than in the raw water, which was caused by the use of PAM-containing coagulant. Pivokonsky’s study showed that water treatment plants can reduce the amount of MNPs by an average of 70–83% (mainly those >50 μm in size) depending on the treatment and purification technologies used [200]. The most abundant polymers in both raw and treated drinking water were PE and PET, followed by PP. Other polymers, such as poly(butyl acrylate) (PBA), polyphenylene sulfide (PPS), polytrimethylene terephthalate (PTT), PS, PMMA, PVC, PAM, PA, PES, and nylon, were also found. What is worth highlighting is that, among the identified compounds, there were 12 different polymers forming the microplastic particles, with PET and PP being detected in significant amounts in both raw and treated water. PS and PMMA were only detected in raw water, and PAM was only present in treated water.

The awareness of the importance of hydration for health leads many people, especially in highly developed countries, to always carry a bottle of drinking water with them. Water is sold in various types of packaging, ranging from glass bottles (rarely used by consumers due to their weight and fragility) to disposable bottles made of PET or recycled PET (rPET), as well as various types of reusable packaging made of metal or plastic (PE, PP, and PA) [17,266]. Polymer analysis of microplastic particles found in water sold in glass, PET, and rPET bottles revealed significant differences [260]. PET was the most abundant polymer (53.4%) in PET bottles, while rPET bottles contained the highest percentage (71.6%) of PE + additives. Glass bottles contained the highest amount of PE particles, with a total of 97.7% (38.2% of normal PE and 59.5% PE + additives), which, according to the authors, originated from the bottle caps. These results are consistent with those of other studies in which PET, PS, and PP, originating from bottles and caps, were the most common polymer types in packaged drinking water [17,266,271,272]. All conclude that bottled water contamination with plastic particles is at least partially caused by the packaging and/or the bottling process itself.

Microplastic particles were also identified in other beverages. Packaged milk from different geographical regions contained various polymers, including PES, PET, PTFE, Nylon 6, PU, PP, PAM, PS, PE, EVA, and PVA, as well as polyethersulfone (PESU) and polysulfone (PSU) [273,274,275,276]. Some of them might be generated from industrial production, dilution, and distribution tools and equipment, such as pipes, valves, and membrane filters used in the dairy industry (PESU, PSU), or from packaging like bottles (PP) and multilayer laminated paper in cartons (PVA, PE). PS, PP, LDPE, HDPE, and PAM were identified in beer [186,277,278], and PET and Nylon 66 were found in tea [149]. PA, poly(ester-amide)s (PEAs), and ABS were found in soft and energy drinks [279]. The polymers detected in wine were mainly natural cellulose, but synthetic MPs belonging mainly to the polyolefin cluster (PE, PP, EVA, and PEVA) were also identified [259]. According to Prata et al. [194], PE probably originates from the synthetic stoppers.

The type of plastic identified in other food products mainly depended on the environmental pollution of soil in which the plants were grown and the animals were bred, or on the level of microplastic contamination in the sea and oceans, when related to fish and seafood. According to calculations made for adult Irish people, the simulated estimated daily intake of MNPs was 1.62 × 10^3^ particles/kg body weight/day. Vegetables (particularly root vegetables) contributed most to MNPs exposure, followed by fruit and grains. This was due to the MNP content in the soil, as well as various bioaccumulation factors, and the consumption of these food types [280]. Using arsenic (As)-containing groundwater in hydroponics enhances the migration of PS microparticles from the environment to carrot roots [39]. Lettuce is one of the most thoroughly studied components of a plant-based diet and is consumed directly or used in various salad mixes. Studies have identified a variety of polymer particles, with PA, PP, PE, HDPE, and LDPE being the most prevalent. Other particles present in lower concentrations included PAM, PVC, PET, CPE, polybutadiene (PB), acrylonitrile butadiene (AB), PC, PMMA, PS, PSU, PU, PVA, and EVA [281,282]. The presence of such a wide range of polymers indicates that MNPs can originate from various sources, including environmental pollution, traffic, pesticides, and fertilisers [282,283]. Since lettuce can absorb contaminants from the soil and water through its root system, as well as from the air and water through its leaves, there is a high risk of MNP contamination. Research by Bai et al. has shown that the highest concentration of MPs is found in the old leaves of leaf lettuce, followed by the new leaves and then the innermost leaves of nodular lettuce [281]. It is important to note that standard washing of lettuce leaves with water alone does not remove all contaminants [284]. Nanoparticles of PVC, PET, PE, and Nylon 66 were detected in samples of cowpea, flowering cabbage, rutabagas, and chieh-qua crops [285]. In Turkey, some common fruits and vegetables (pears, apples, tomatoes, onions, potatoes, and cucumbers) were analysed, revealing the three main polymers to be LDPE, PP, and PET [213].

The main polymers identified in MPs isolated from Indian rice were PE, PET, PP, and PA [286]. Rajendran et al. [287] analysed MPs from various edible plants. They identified nylon particles in potatoes and grapes; nylon, high-density polyethylene terephthalate (HDPET), polyethylene oxide (PEO), and PS in bananas; and in brinjal—ABS, nylon, and PEO were present. As many as 12 different types of polymer were identified in samples of refined wheat flour from Bangladesh: ABS, EVA, HDPE, LDPE, latex, nitrile, nylon, PC, PET, PMMA, PP, and PS [288]. The average number of MPs ranged from 2747 to 6409 per kilogram of branded and non-branded flour, respectively, and they mainly originated from the flour packaging.

Tests on food of animal origin revealed the presence of PMMA in 68% of Dutch cows’ milk samples and polymers of polyvinyl chloride (PVC-P) in 16%, while PP and polymers of styrene (Styr-P) were not detected in any of the milk samples [289]. Conversely, PE was found in the Dutch meat samples, with a higher concentration in the beef than in the pork. Other polymers that were identified in some meat samples included PVC-P and Styr-P, while PMMA, PP, and PET were not detected in meat. In another study, the following substances were detected in chicken carcasses (in descending order): PVC, LDPE, PS, and PP homopolymer [290].

When analysing studies on polymers detected in various milk samples, a clear difference in composition was observed between milk samples from different countries. Samples of pasteurised, packaged milk purchased from local markets and stores in India were found to contain PE, PP, and PAM [276]. In samples of milk sold in Switzerland, including raw milk, whole liquid milk, skimmed liquid milk, and skimmed milk powder, PE, PES, PP, PTFE, and PS were detected, as well as smaller amounts of PA, PU, PSU, and PVA [274]. Thermoplastic sulfone polymers (PESU and PSU) were detected in all 23 samples of packaged milk of various types (for adults and children; whole milk, half-fat milk, light/low-fat milk, lactose-free and lactose-free light milk; pasteurised milk and ultra-high-temperature processed milk) purchased from supermarkets, local stores, and pharmacies in Mexico [275]. PESU, which is used for membrane materials in various dairy processes, was the polymer most frequently detected in this study. Basaran et al. [273] examined samples of UHT cow’s milk (including lactose-free and toddler milk) packaged in paper, cardboard, plastic, and aluminium, which were purchased in Turkish shops. The authors identified five different types of polymer, including Nylon 6, PET, EVA, PP, and PU, with EVA and Nylon 6 accounting for over 85% of the polymers. Interestingly, PE, the polymer used in packaging in this study, was not detected, suggesting that the MPs found in the milk samples could not have originated from the packaging. These results demonstrate that the quality of raw milk, influenced by environmental, water, and feed pollution, as well as the processes, machines, and materials used in the production of packaged milk, significantly impacts the number of MNPs detected and their chemical composition [274].

An analysis of commonly consumed protein-rich products in the USA, including both animal-based products such as shrimp, chicken breast and nuggets, sirloin steak, pork chops, and minced pollock, as well as alternative plant-based products such as vegetable nuggets, vegetable fish sticks, minced vegetable beef, and tofu, revealed the presence of PET/PES, PE, and PP [291]. Similar polymers have been identified in protein- and carbohydrate-rich feed ingredients commonly used for aquatic animals, which may pose a risk to them and thus enter the food chain. The dominant polymers detected in fish meal, soybean meal, poultry by-products, rice bran, wheat bran, and wheat flour were PP, followed by LDPE, PS, PE, nylon, ABS, PES, RY, PET, and CP [292]. In the tested fish and shrimp meals, the main chemical components of MPs isolated from the samples were olefins and PES (for fibrous MPs) and paraffin and PE (for film-like and fragmentary MPs) [293]. Nalbone et al. detected several plastic polymers in fresh and processed mussels sampled from the Italian market, with PE being the dominant type [210]. Fish, both fresh, canned, and dried, similarly to other seafood (crab, shrimps, prawns, squid, crabs, etc.), contain mainly PE, PET, PS, PVC, PP, and PA [29,294,295]. Among the various polymers detected in edible seaweed and nori produced from it were PE, RY, PP, PET, CP, and PS [296,297], while in benthic organisms, PA, PE, PET, and CP dominated [298]. Among 121 MPs detected in oysters, the most abundant was PET, accounting for 34%, followed by PP, PES, PS, CP, PVC, PA, and EPS [299]. PE and PET have also been identified in *Donax trunculus* (truncate donax, wedge clams) collected from Class A production areas in the Tyrrhenian Sea (Mediterranean Sea) and therefore destined directly for the final consumer [300]. The most abundant polymers in canned sardines and sprats were PP and PET [301]. According to other studies, the main polymers identified are PE, PA, CMFs, PP, PET, PVC, and RY in mussels; PE, PET, and PES in wedge clams; and PP, PE, PES, and PVC in oysters [294,302].

Some microplastic particles (identified as PE) were detected in eggs, and their level was higher in egg yolks than in egg whites, probably due to their lipophilic nature [303].

A high number of microplastic particles, mainly smaller than 10 μm, were detected in four types of commercially available edible oil bottled in PET (olive oil, rapeseed oil, sunflower oil, and coconut oil). The dominant polymers identified were poly(ethyl acrylate:St:acrylamide) and poly(ethylene:propylene); however, no PET microplastics were detected in the oils after incubation for 10 days at 40 °C [304]. MPs in the form of fragments measuring less than 100 μm and consisting primarily of PE (50.3%) and PP (28.7%) were found in edible vegetable oils sold commercially in Italy and Spain. These oils included extra-virgin olive oil, olive oil, sunflower oil, and a mixture of oils [305].

Four different polymers were identified in samples of honey produced in Turkey: PE (62%) and Nylon 6 (22.3%), followed by EVA and PP [306]. Other studies on honey have also detected PET, PA, LDPE, HDPE, polycaprolactone (PCL), polyvinyl stearate (PVS), RY, PTFE, EP, ABS, and PAM [307,308]. Contamination with plastic can be caused directly by honeybees, which incorporate MP particles from food and transfer them to honey, or by technological and packaging processes [308,309,310].

Although salt and sugar are just the additives used in food, many people use them in large quantities that far exceed nutritional guidelines. The type of plastic polymers found in salt depends on its origin. Table salt can be produced from sea salt, so the amount and type of MNPs will be influenced by the geological location of origin, as well as subsequent stages of production (e.g., solar evaporation, crystallisation, packaging, and transport). Rock and well salts can also become contaminated during collection and drying. Refined salt can become polluted during mechanical treatment, while an unrefined type of salt is exposed to MNP contamination from the air on the surface of the crystallisers [311]. The main types of plastic identified in salt are PE, PP, PET, and, to a lesser extent, PES, polyalkene (PAK), PU, PEVA, EVA, PS, PVC, PA6, CP, and others [188,216,312,313,314]. Only PE was detected in sugar brands [216], while PE and HDPE were found as the most likely polymers in bottled vinegar [315].

It is worth noting that, in addition to the MNPs found in salt, sugar, and food seasonings, further polymer particles may be abraded onto food when mechanical grinders are used for salt or spice disruption. The chemical composition of these particles depends on the materials and mechanisms employed by the grinder manufacturer. Research from Poland [316] showed that both coarse pink Himalayan salt and whole black peppercorns significantly abrade the polymeric parts of grinder burrs. Consequently, significant quantities of PET, POM, PS, and PMMA were released in proportion to the length of time the mechanism was used, and the extent to which the plastic grinding cutters were worn down.

### 4.3. Plastic-Associated Chemicals and Other Contaminants

It should be remembered that various contaminants can also enter the body alongside MNPs and negatively influence living organisms. These include additives used in the production of plastics, such as plasticisers (e.g., phthalates, bisphenol A), fillers, flame retardants, heat or light stabilisers, colourants, antioxidants, lubricants, and antistatic agents [15,317], which also have undesirable or even toxic effects [35,40,41,318,319]. In addition, when MNPs move through an aquatic environment or soil, they can absorb (reversibly or irreversibly) various pollutants [317,320,321], including various persistent organic pollutants (POPs), such as polycyclic aromatic hydrocarbons (PAHs), polychlorinated biphenyls (PCBs), and organochlorine pesticides (OCPs), as well as polybrominated diphenyl ethers (PBDEs), aliphatic hydrocarbons, bisphenol A (BPA), perfluorinated chemicals (PFCs)/perfluoroalkyl acids (PFAAs). Other contaminants adsorbed by MNPs include heavy metals (aluminium, arsenic, cadmium, chromium, cobalt, copper, iron, lead, manganese, nickel, titanium, and zinc) [321,322,323], toxins [317,324], antibiotics [325,326], various pharmaceuticals [135,327,328], pollens and allergens [329,330], pesticides [331,332], herbicides [333], viruses [334] and pathogenic microorganisms [335,336,337,338]. These compounds and microorganisms can then be released into the environment or food chain [339,340], or upon inhalation/ingestion of microplastic particles, they can enter organisms and exert harmful effects [341,342,343,344,345,346,347]. These effects include poisoning [321], disorders of physiological processes, the spreading of infectious diseases [348], or the increase in antibiotic resistance in microorganisms [317,331,349,350].

Triclosan (2,4,4′-trichloro-2′-hydroxydiphenyl ether, TCS) is a chemical compound with antimicrobial activity that was often added to food storage containers and kitchen utensils. It has been shown that TCS can exert a negative impact on human health. Consequently, it has been removed from the list of authorised additives in some countries, e.g., the EU; however, many food containers containing TCS are still available through online sales platforms [351]. It has also been proven that MPs are released from these containers and that TCS migrates to food simulants when the containers are heated in conventional and microwave ovens.

## 5. Food Pyramid as a Graphical Representation of FDBG and Various Dietary Patterns

The food pyramid was first introduced by the Swedes in 1974. Its creation was linked to high food prices and the person of Anna-Britt Agnsäter, who wanted to show the Swedish population how to eat healthily and reduce their fat intake. She first published her concept in the Vi magazine. Her pyramid had cheap and nutritious ‘basic products’ at the bottom, including milk, cheese, margarine, bread, potatoes, and cereals, followed by a large section of fruit and vegetables. At the top were ‘supplementary products’ such as meat, fish, and eggs, which were intended to provide nutrients that were insufficient in the basic products [352]. Around the same time, proposals appeared for a graphical presentation of the recommendations in the form of a ‘food wheel’. Although the food wheel still appears occasionally in the recommendations from various institutions, critics point out that this graphical form does not accurately reflect the proportions of a given product group in the daily diet. A proper graphical presentation should take into account not only what to eat, but also the quantities and frequency (e.g., daily, several times a week, or a month). It could also suggest the recommended number of meals per day and the amount of water to drink, as well as the level of physical activity required. Poland’s food pyramid, proposed by the Food and Nutrition Institute in 2016 [353], is a good example of this. It presents a list of recommended food quantities and frequencies, and also takes lifestyle factors such as physical activity and sleep into account, as well as ‘non-nutritional’ aspects such as environmental and social issues.

Graphical representations of dietary recommendations for consumers are common. The pyramid, in particular, immediately shows what constitutes the basis of a diet, i.e., which foods should be consumed in the largest quantities and most frequently, while at the top of the triangle are foods that people should only consume occasionally. The pyramid concept has gained many supporters and has been adopted in other countries, where it has been adapted to local conditions, including climatic and geographical ones. Various food pyramids are currently in use in countries and regions such as Scandinavia, Sri Lanka, the United States, China, and Japan. Food pyramids and other forms of food-based dietary guidelines (FBDGs) have also been presented on the websites of various European and international institutions such as the WHO, the FAO, the USDA, the EFSA, and the EU [354]. When analysing FBDGs in different parts of the world, it becomes apparent that they reflect not only cultural differences or the dominant religions in the region (e.g., the prohibition of alcohol and pork), but also the types of food available (different types of cereals, fruits, or vegetables). In some cases, they also reflect fundamentally different dietary recommendations or more common deficiencies (e.g., high lactose or cow’s milk intolerance leads to its replacement with yoghurt and goat’s milk products in the Mediterranean region or soy in Asia).

Almost every European country has its own dietary recommendations, which are often presented in a unique graphic format that continues to evolve as new knowledge about the diet’s impact on health becomes available [355]. Throughout the history of European ‘nutritional guidelines’, various food pyramids have appeared alongside alternative graphical presentations, such as the Hungarian ‘house’, the French ‘stairs’, the Swedish ‘traffic lights’, and the English ‘Eatwell plate’. Other continents also have their own pyramids and variations, including the pagoda in China, the fan in Bolivia, and the four-leaf clover in Turkey [356,357,358,359].

There are also food pyramids designed for specific population groups within a country, such as adults, children, the elderly, pregnant women, and people with certain conditions (e.g., the Crohn’s Disease Food Pyramid or the Cardiac Food Pyramid). The basic adult version of the pyramid differs from those for older people and children, of course. This is due to the specific nutritional needs of seniors and the challenges associated with children’s intensive growth and development, including physical challenges. Furthermore, given the limited access to animal protein in many parts of the world, and the growing awareness of the environmental and health benefits of replacing meat with plant-based products in other regions, versions of the food pyramid for vegetarians and vegans have also emerged. Depending on the cereals, fruits, and vegetables available in a given geographical region, different types of vegetarian diets can be identified, and therefore different pyramids can be created. When establishing a vegetarian food pyramid, it is important to bear in mind that, unlike meat, plants only occasionally contain complete proteins, i.e., proteins containing all the essential amino acids. Therefore, in order to obtain a complete protein profile, it is necessary to either consume a variety of protein-rich vegetarian products (known as complementary protein building) or include plant products containing all nine essential amino acids in the diet. Currently, the term ‘traditional vegetarian diet’ usually refers to the healthy, traditional, lacto-ovo-vegetarian diets practised in Europe, North America, South America, and Asia. These are typically anchored on the Mediterranean diet pyramid, which is widely regarded as one of the healthiest.

In recent years, Europe has experienced an increase in the prevalence of non-communicable diseases such as cardiovascular disease, type 2 diabetes, hypertension, obesity, and various cancers [360]. In response to this issue, numerous institutions are attempting to define appropriate nutritional targets for the population in the form of coherent and understandable national FBDGs. These FBDGs should form the basis for preventing non-communicable diseases and be tailored to the specific needs of a country and its inhabitants. Crucially, they must also be consistent with policies that promote food safety, a healthy environment, and the local food economy. It should be noted that dietary recommendations often differ significantly from the actual consumption of various nutrients, particularly in unhealthy diets, which are commonly followed by young and hard-working people in highly developed countries. The best example of this is the so-called Western diet (also known as the Standard American Diet (SAD)). This dietary pattern is characterised by a high consumption of ultra-processed foods, such as French fries, burgers, pizza, and takeaway meals, as well as coffee, energy drinks, and sweetened beverages. At the same time, there is low consumption of fruit, vegetables, and dairy products. This results in an increased intake of saturated fats, refined carbohydrates, and salt. In addition, these meals are often not prepared at home, but bought as takeaways or delivered by catering companies in disposable plastic packaging (so-called ‘box diets’). According to the Food and Agriculture Organization of the United Nations (FAOSTAT), the average food supply for Americans (a nation that commonly uses the SAD) increased from 3754 kcal/capita/day in 2015 to 3912 kcal/capita/day in 2022 [361].

According to the WHO, 25 of its European Region member states already have government-approved FBDGs, and eight countries are developing such guidelines [362]. Taking into account climatic conditions, historical factors, and the traditions of crop cultivation and animal husbandry that determine the most readily available food raw materials, as well as typical patterns of food consumption, European countries can be divided into four groups (the National FBDGs applicable to selected representatives of a given region are presented in the Appendix A [355,362,363]):Western Europe (WE): Austria, Belgium, France, Germany, Ireland, Luxembourg, Liechtenstein, the Netherlands, Switzerland, the United Kingdom (Appendix A in Appendix A);Nordic and Baltic countries (NBC): Denmark, Finland, Sweden, Iceland, Norway, Estonia, Latvia, and Lithuania (Appendix A);Central and Eastern Europe (CEE): Belarus, Bulgaria, Czech Republic, Hungary, Poland, Romania, Slovakia, Serbia, North Macedonia, Moldova, Slovenia, Bosnia and Herzegovina, Ukraine (Appendix A);Southern Europe (SE): Albania, Andorra, Greece, Italy, Malta, Monaco, Portugal, Spain, Croatia, Montenegro, San Marino, and Turkey (Appendix A).

## 6. Assessment of the MNP Intake Depending on Dietary Patterns

It is obvious that the human diet must provide nutrients, primarily protein, fat, and carbohydrates, which are essential for survival and good health. These basic nutrients originate from completely different raw materials and foodstuffs, depending on dietary habits and geographical location, which means they provide different amounts of MNP particles.

In many countries, especially poorer ones, fish caught by people themselves is the most important source of protein. The same applies to commercially available seafood [294], regardless of whether it is fresh or canned, as there is a risk of inadvertently consuming MNPs alongside it [210,364,365,366]. The content of MNPs in edible parts is usually lower than in inedible parts [367]. Many types of seafood, such as mussels [212,368], prawns [369], squid [370], sardines [371], and sprats [301], are consumed in their entirety, thereby increasing the likelihood of ingesting the microplastic particles present in their digestive tracts. This is because these organisms are filter feeders that unintentionally ingest MNPs along with their food, which then remain present in their digestive tracts [372]. However, numerous studies show that the edible parts of fish and seafood also contain MNPs [207,209,295]. In countries without easy access to fish, legumes are often a source of protein. There are increasingly frequent reports of MPs being absorbed by plant roots from soil or water, or by leaves through stomata from water or air. These MPs are then transported to other parts of the plant [36]. The MNPs have been detected in various plants [373], including protein-rich crops such as beans [374], peas [37,375], and *Vicia faba* [38]. In more industrialised regions, animal products such as eggs, milk, and meat are a key source of protein. The consumption of these products obviously depends on geographical conditions, including the types of animals typical of a given area and the climate, as well as the dominant religion in a given country (e.g., forbidding the consumption of pork). MPs have been detected in all of these products, and their concentration depends on the type of raw material used, including its origin, species, breed, and feed, as well as the degree of processing and storage (e.g., packaging) [228,276,290,291].

Carbohydrates can be found in a wide variety of foods. Healthy sources include unprocessed or minimally processed whole grains, vegetables, fruits, and beans, while unhealthy sources include white bread, rice, pastries, sodas, and other highly processed or refined foods. The recommended intake for adults and children over 1 year old is about 130 g of carbohydrates per day, which should account for 45–65% of total calories [376]. This means that carbohydrates have (or at least should have) the largest share in the diet and, depending on consumer choices and the products consumed, can have completely different effects on health, including the amount of MNPs ingested [213,377,378]. It should also be remembered that corn, oats, barley, wheat, and sorghum serve as the basis for animal feed [379]. The presence of MNPs in these plants can result in the transfer of MNPs to animals and, indirectly, to humans. Studies have shown that the ability of different fruits and vegetables to absorb MNPs from the environment varies, as does the part of the plant where MNPs accumulate (which is not always the part consumed by humans) [36,280,380]. However, the amount of MNPs in individual plant parts also depends on soil contamination levels and the agronomic systems, fertilisers, and irrigation methods used [381,382]. The potential risk of ingesting MNPs is higher for processed foods [383], as well as products containing added sugar or honey, which may also contain MNPs [216,384,385]. Domenech and Marcos [220] stated that the mean number of microplastic particles ingested with a daily portion of fruit and vegetables (400 g) accounts for 19.38 × 10^9^ particles per year. This is higher than the MNPs number ingested through seafood, which was estimated at 22.04 × 10^3^ particles per year.

Fats in the human diet can come from animals (both land and aquatic species) or plant sources (extracted from oilseeds or found in edible fruits, seeds, and kernels). The first group includes butter, lard, and pork fat, as well as foods that are rich in fat, such as meat, fish, eggs, and dairy products (including various cheeses). Considered healthier, vegetable fats include olive oil, rapeseed oil, coconut oil, and palm oil. Other sources of plant lipids include walnuts, almonds, avocados, olives, sunflower seeds, pumpkin seeds, and sesame seeds. As mentioned earlier, fish, meat, milk, and dairy products can be contaminated with MPs [228,256,273,386,387], but vegetable fats are also not free from MNPs [304,305].

### 6.1. Methodology Used for the Assessment

Based on the available data on nutritional recommendations in European countries mentioned in the previous chapter, food products can be divided into seven categories that are most commonly included in the food pyramid: (1) water; (2) grains (bread, rice, pasta, cereals) and potatoes; (3) vegetables and fruit, including juices; (4) milk and dairy products; (5) protein sources (meat, fish, eggs, legumes); (6) oils, fats, nuts, and seeds; and (7) culinary additives to food (salt, sugar, honey, spices). Due to significant differences in the recommendations formulated within the FBDGs between subregions and countries, it is impossible to estimate the amount of MNPs consumed by the European population. Moreover, significant differences in the amount of MNPs can be noted within one food category. For example, differences have been observed between fish/seafood and poultry, and between sweetened and unsweetened beverages. For the purposes of this review, therefore, three artificial ‘average food pyramids’ or dietary patterns were constructed in order to estimate MNPs consumption. All of these pyramids contain the items listed above, but—due to the aforementioned reasons—food products were categorised into 12 groups. Firstly, fruit was separated from vegetables, and potatoes were included in the latter category. Secondly, fish were considered separately from meat, which can include poultry and red meat, as well as highly processed meat, depending on the dietary pattern. In many FBDGs, red meat and processed meat are recommended to be replaced with legume seeds (e.g., beans, peas, chickpeas, lentils, and broad beans) and nuts. Therefore, we created a separate food group, number 6, for legumes and nuts. Moreover, water was added as a separate group, because adequate hydration is the basis of proper nutrition and a healthy lifestyle in all recommendations. However, depending on the dietary pattern adopted, this can be achieved by drinking pure tap water (the recommended option), as well as various types of bottled beverages, including sweetened ones. As the number of MNPs varies according to the type of water, average values for tap water and pure water packaged in plastic bottles (in a 1:1 ratio) were used in the calculations. For the Western diet only, bottled water and sweetened beverages (in a 1:1 ratio) were used as the fluids for hydration. Although all 12 food groups are mentioned in the tables presenting the dietary patterns, their proportions vary depending on the chosen diet. In some cases, the values equal zero when a particular food is not consumed (e.g., meat in a lacto-ovo-vegetarian diet).

For the calculation, we generally used the relationship whereby 1 g of either sugar or protein provides 4 kcal of energy, whereas 1 g of fat provides 9 kcal. However, the average caloric values of individual product groups were estimated primarily based on available literature. In each pyramid (dietary pattern), we took a calorie requirement of 2000 kcal into account, but this was fulfilled with different proportions of food products.

None of the proposed pyramids included alcohol, such as wine and beer, as the decision to consume or not consume alcohol is more an individual choice than a cultural or dietary one.

The average number of MNPs in a given category of food was estimated based on the available literature. All available databases, platforms, and search engines were utilised in the search for publications and other relevant documents. These included, for example, PubMed, Web of Science, Scopus, Science Direct, Google Scholar, ResearchGate, Academia, JSTOR, DOAJ, EBSCO, AGRICOLA, Embase, Archive.org, Science.gov, and Refseek. Searches were also conducted on the websites of institutions such as the WHO, FAO, and EFSA, as well as on the websites of individual journals and publishers (e.g., Oxford Academic Journals, Elsevier, Springer Link, RSC Journals, MDPI, ACS Journals, AIP Journals, APS Journals, SAGE Journals, De Gruyter, Taylor & Francis, and Wiley Online Library). As the most popular databases did not yield enough useful articles for estimation purposes, a manual search was conducted. This involved entering the desired detailed keywords, together with the names of particular European countries and different types of fruit, vegetables, cereals, seafood, and meat, into various databases. This search was conducted using various combinations of the following keywords: particle(s), microparticle(s), plastic, microplastic(s), nanoplastic(s), MP, MPs, NP, NPs, MNP, MNPs, microfragment(s), film(s), microfilm(s), fibre(s), microfibre(s), polymer(s), microbead(s), food, food products, foodstuff, processed food, ultra-processed food, refined food, food preparation, fast food, ready-to-eat products, consumption, ingestion, intake, level, concentration, number, amount, trophic transfer, trophic chain, food chain, storage, packaging, disposable, reusable, take-out, takeaway, container(s), box(es), bottle(s), dish(es), cap(s), teabag(s), capsule(s), contamination, pollution, water, tap water, treated water, potable water, drinkable water, juice, beer, wine, drink(s), beverage(s), mineral water, groundwater, coffee, tea, soft drink, cola, plant(s), edible parts, root, leaves, fruit(s), vegetable(s), avocados, olives, banana, berry, berries, apple, pear, orange, carrot, lettuce, broccoli, onion, tomato, cucumber, potato, tofu, nori, legume(s), soybean, soy, bean(s), peas, chickpeas, lentils, broad beans, corn, oats, barley, wheat, sorghum, rye, grains, whole grains, cereals, flour, bread, rice, pasta, pastries, salt, sugar, honey, seasonings, spices, French fries, animal(s), animal products, dairy, milk, cheese, yoghurt, egg(s), pork, turkey, chicken, poultry, meat, red meat, butter, lard, steak, beef, sausages, bacon, seafood, oysters, crabs, mussels, prawns, shrimp, squid, fish, sardine(s), sprat(s), bivalves, crustaceous, burgers, oil(s), fats, olive oil, rapeseed oil, coconut oil, palm oil, sunflower oil, oilseed, seed(s), kernels, sunflower seeds, pumpkin seeds, sesame seeds, nuts, walnuts, almonds.

The main criteria for selecting literature sources to estimate MNP concentration in food and MNP intake were (a) access to the full text of the publication; (b) data available in English or Polish; (c) results relating to a European country (e.g., MNP intake in Europe, plant and animal raw materials and water originating from Europe); and (d) the publication provided the number of MNPs in particles/g or particles/cm^3^. The results of the search are presented in Table 1. Where studies on raw materials and food products originating from the European market were available, the averages were calculated based on these values (e.g., fruit, vegetables). For product groups for which no such results have been published, data from other continents was used or, in extreme cases, data from review papers (without access to source data).

Due to the enormous variation in the abundance of MNPs reported by various researchers, the arithmetic mean was used to estimate MNP intake. This was calculated using data from all available studies analysing food from a given group. For example, the MNP content in vegetables has been calculated based on the average number of MNPs reported in carrots, lettuce, broccoli, onions, tomatoes, and potatoes, whereas the MNP content in fruit was calculated based on the presence of these particles in apples and pears. The literature sources used in the estimation have also been listed in Table 1.

Only one publication reporting on the MNP content of eggs was found. For the purposes of the calculations, it was assumed that the average egg weighed 55 g and the microplastic content present in 1 g of egg was then calculated.

In order to calculate how many MNPs are consumed with food from group 9b (meat and ultra-processed food), depending on the dietary pattern used, two assumptions were made.

(1) In the case of ‘take-out’ meals, representing processed solid food, data from a single publication [251] was used for the calculations. This publication presents the average MNP content in various products sold in takeaway containers. Other available publications have reported significantly higher values (by several orders of magnitude), but these should be treated with caution. These differences depend on various factors, including the material used for the packaging, the type of food packaged (e.g., acidic, aqueous, or carbonated), and the temperature (e.g., freezing, cooling, room temperature, pouring hot water, or heating in a microwave). However, these publications express the number of MNPs released into food as ‘particles per cm^2^ of packaging’ [148,230]. This makes it difficult to convert the results and express them as ‘particles per gram of food’. This is because the number depends on the type of food (e.g., acidic, aqueous, liquid, or solid), its consistency, the size of the portion in the packaging, and whether the food was consumed immediately or transported, shaken, or heated. Therefore, it should be acknowledged that the value used in the calculations significantly underestimates the actual number of MNPs present in packaged takeaway food.

(2) As an example of processed liquid food, cold beverages, coffee, and tea, purchased in plastic cups for takeaway, have been chosen. In this case, again, the differences in the reported number of MNPs released into the beverage were enormous and depended on many factors. For the calculations, a publication that used a method most closely resembling the typical brewing of tea (i.e., 250 cm^3^ of deionised water was added at a temperature of 95 °C for 5 min) [397] was selected. Other publications in which the number of MNP particles released from one bag reached billions of particles (up to 11.6 × 10^9^) [149] were excluded from calculations, due to a completely different methodology of determination being employed.

Based on the assumptions above, the amount of MNPs in 1 g of food from group 9b was estimated using the following proportions: 1 portion of meat (200 g): 1 takeaway meal (400 g): 1 takeaway beverage (250 cm^3^ portion).

Generally, the average concentration of MNPs in a given food group was adopted for the further part of our review (Section 6.2). However, it should be noted that the differences presented in the available literature are enormous. This is primarily due to the type of raw material itself, its species or variety, and which part of the raw material is edible (e.g., whether it is a root, leaf, flower, fruit, or seed in the case of plant raw material). In addition, the level of MNPs depends on the cultivation method and agronomic measures employed (protective film, fertilisers, etc.), the contamination of the cultivated soil and the water used for irrigation, and the general environmental pollution at the place of cultivation. In the case of products of animal origin, differences result from the conditions of rearing, which are particularly important for aquatic animals (water pollution), as well as the type of animal (its diet and place in the food chain). Other factors include the raw material consumed (whether it is milk, a whole animal, or only selected parts, e.g., muscles), the type of feed used, and the method of processing before testing (packaging, transport, heat treatment).

The analytical methods used also have a huge impact on the values obtained. Some of them are limited to testing microplastics only; others test microplastics and nanoplastics separately, and still others test all of these particles. Furthermore, specific methods have limitations regarding the size of particles that can be detected; some are capable of detecting particles as small as nanometres, while others can only detect particles measured in micrometres or millimetres. Moreover, studies present data in various units such as particles/individual, particles/g, particles/kg, particles/dm^3^, μg of plastic/g of food, and so on.

### 6.2. MNP Intake According to Three Different Dietary Patterns in Europe

The aim of this review was to estimate the effect of the adopted dietary pattern on the amount of MNPs consumption; therefore, the same average values (from Table 1) were used for all pyramids and dietary patterns in the calculations. This implies that the MNPs level in a given food group (1–12) remains constant, while the proportions of individual dietary components within the total daily diet vary.

The pyramids that have been designed represent the three most important dietary patterns in Europe:

Pattern 1—a Mediterranean diet, or more specifically, the healthy eating pyramid based on the Mediterranean diet (Figure 4), which, according to FBDGs formulated by the WHO for the European region [362], is rich in vegetables and fruit, with fish and seafood as the main source of protein and olive oil as the main source of lipids; it is common for SE and NBC (Table 2);

Pattern 2—an unhealthy, but unfortunately common in WE and SSE, Western diet pyramid (Figure 5), which often includes highly processed foods (e.g., white bread, refined grains, ready-to-eat products, takeaway food and drinks) and significant amounts of red meat, as well as carbonated drinks, sweets, salted snacks, and food additives (sugar, salt) (Table 3);

Pattern 3—a lacto-ovo-vegetarian pyramid (Figure 6), an increasingly popular alternative to the vegetarian diet, which has ideological roots and is primarily selected by people who do not want to contribute to violence against farm animals, while at the same time wanting to lead a healthy lifestyle and take responsibility for the natural environment (Table 4).

These diets not only contain different proportions of protein, fat, and carbohydrate sources but also different sources of these nutrients. Therefore, a diet that is beneficial to human health due to the nutrients it provides may be detrimental due to the MNPs contained in the raw materials and products used.

In the healthy eating dietary pattern (Pattern 1), fish, seafood, legumes, and fresh dairy products are the main sources of protein, while carbohydrates and dietary fibre are provided by pasta, rice, and bread made from whole-grain cereals, together with vegetables and fruits. Extra-virgin olive oil and nuts provide healthy oils [403]. The recommendations also include the need for adequate hydration (~2 dm^3^ of water per day), preferably using the public water supply (tap water); however bottled water consumption has also been included. This dietary pattern also advises limiting added sugars and salt, as these additives have been linked to chronic diseases (Figure 4, Table 2). However, as mentioned above, in addition to salt and sugar, fish, seafood, vegetables, and tap water may also contain high levels of microparticles.

**Figure 4 molecules-30-03666-f004:**
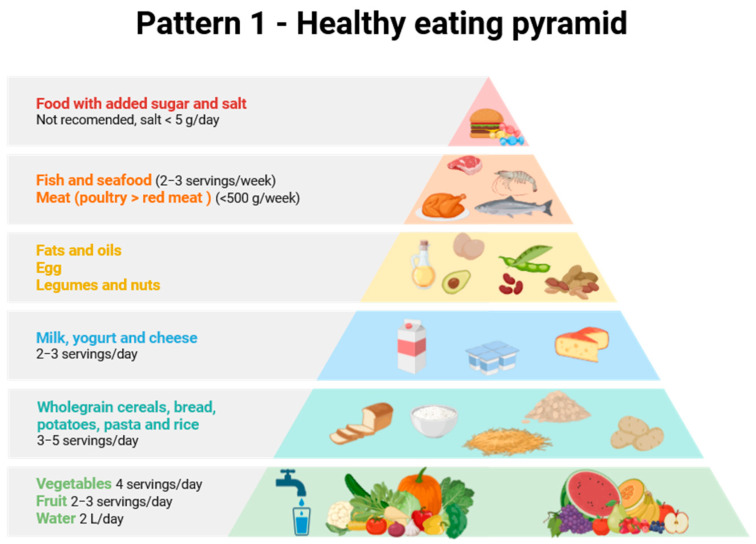
Food pyramid representing a healthy eating dietary pattern based on the Mediterranean diet. Graphic prepared in BioRender [119].

**Table 2 molecules-30-03666-t002:** The healthy eating dietary pattern (Pattern 1) based on the Mediterranean diet adapted to 2000 kcal [403,404,405,406,407].

No	Food Group	Daily Amount	Estimated Energy (kcal)	Energy Percentage (%)	Justification/Comments
1	Vegetables	400 g	100	5.0	Minimum 4 servings of raw and cooked vegetables per day, which have high volume, but low calories.
2	Fruit	200 g	90	4.5	2–3 servings (½ cup per portion) per day of fresh, preferably seasonal, colourful fruits, mainly berries.
3	Cereals and grain products	250 g	825	41.2	Main source of energy included whole-grain bread, rice, and pasta.
4	Fats	35 g	309	15.5	The fats consumed are olive oil and other vegetable oils that are rich in monounsaturated fatty acids.
5	Fish and seafood	35 g	100	5.0	It is a source of high-quality protein: 2–3 servings (200–300 g) per week are recommended.
6	Legumes and nuts	30 g + 20 g	90 + 100	9.5	A source of protein and fibre, as well as healthy fats from nuts, is recommended at 2–3 servings per day.
7	Dairy	300 g	225	11.2	It is a source of calcium and protein: 2–3 servings per day, mainly milk, yoghurt, curd, and ripened cheese, but for ripened cheeses, only one slice per day is recommended.
8	Eggs	55 g	75	3.8	Few eggs per week are recommended, including those in pasta or bread.
9a	Meat	43 g	86	4.3	Limited consumption is recommended several times a week (no more than 500 g/week), with poultry and lean meat being preferable to red meat and processed meat.
10a	Water (tap water)	1000 dm^3^	0	0	No energy, but a carrier of MNPs.
10b	Water (bottled)	1000 dm^3^	0	0	No energy, but a carrier of MNPs.
11	Salt	5 g	0	0	The recommended amount is less than 5 g of salt (2 g of Na) per day.
12	Sugar	0 g	0	0	There is no added sugar in the diet; only carbohydrates that are present naturally in the raw materials.

The Western dietary pattern (Pattern 2) has been based on the SAD adapted to European conditions [408,409]. It is rich in processed and ultra-processed foods (UPF) and saturated fats, mainly of animal origin. UPFs such as fast food, takeaways, and ready meals, together with meat (mainly red meat and processed meat), can account for 30–60% of energy intake, while fats are responsible for supplying up to 40% (Figure 5, Table 3).

Moreover, people who follow this dietary pattern often replace water and fresh fruit with fruit juices, carbonated drinks, energy drinks, takeaway beverages, coffee, or tea. They also tend to eat large quantities of snacks and sweets. Taken together, these meals and beverages can constitute a significant source of MNPs, as well as sugar and calories in a diet. Milne et al. [291] reported that the number of MNP particles in protein sources increases with the level of food processing. The estimated total amount of sugar provided by this dietary pattern easily exceeds the WHO recommendations (up to 10% of energy).

**Figure 5 molecules-30-03666-f005:**
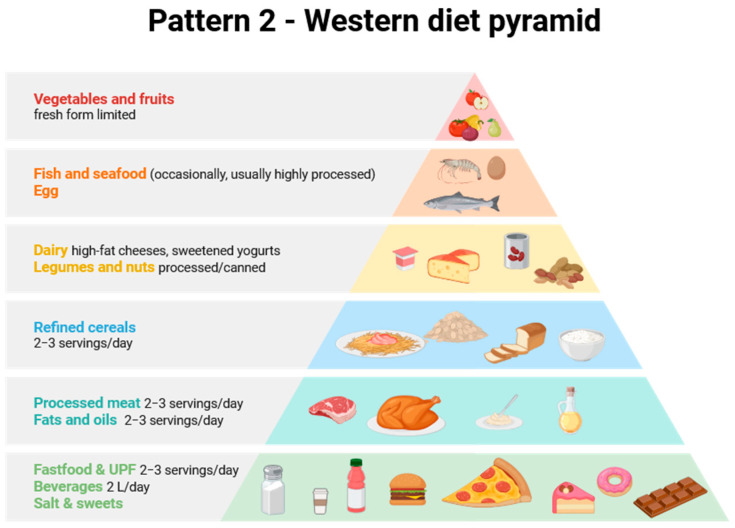
Food pyramid representing a Western dietary pattern. Graphic prepared in BioRender [119].

**Table 3 molecules-30-03666-t003:** The Western dietary pattern (Pattern 2) adapted to 2000 kcal [406,408,410].

No	Food Group	Daily Amount	Estimated Energy (kcal)	Energy Percentage (%)	Justification/Comments
1	Vegetables	50 g	12.5	0.6	Very small quantities, often negligible, consumption significantly below recommendations.
2	Fruit	50 g	22.5	1.1	Very small quantities, often negligible, provided mainly in the form of processed fruit (canned or dried). Fruit juices have been included in group 12.
3	Cereals and grain products	80 g	160	8.0	Mainly in the form of refined grains, white (wheat) rolls/bread, white rice, pasta, and sweet breakfast cereals.
4	Fats	70 g	560	28.0	The fats consumed are mainly butter, animal fats, frying fat, and palm oil, which are rich in saturated and trans fatty acids. Can provide up to 40% of energy.
5	Fish and seafood	6,4	18	0.9	Very small quantities, often negligible in daily diet. Often in the form of canned fish.
6	Legumes and nuts	10 g + 10 g	30 + 50	4.0	Mainly in the form of processed foods (canned) and sweet/salted snacks.
7	Dairy	40 g	140	7.0	Mainly high-fat melt-type cheese, cream, ready-made sweetened yoghurt, or milk powder. It is a source of calcium, protein, and fat.
8	Eggs	25 g	35	1.8	Average consumption of ½ egg per day, usually eaten in processed foods or breakfast.
9b	Meat and UPF	260 g	650	32.5	High consumption of red meat and processed meats (e.g., sausages and bacon), as well as fast food, takeaways, and ready meals. This is a significant source of fat and salt. Meat and UPFs can provide up to 60% of energy.
10a	Water (tap water)	1000 dm^3^	0	0	No energy, but a carrier of MNPs.
10c	Bottled beverages	1000 dm^3^	0	0	Includes fruit juices, carbonated and energy drinks, takeaway coffee and tea, sugar-free and sweetened (sugar was reported in group 12).
11	Salt	10 g	0	0	High consumption as a result of processed food, salted snacks, and meat consumption.
12	Sugar	80.5 g	322	16.1	Sugar added to sweets, energy snacks, chocolate bars, ice cream, and additives.

The lacto-ovo-vegetarian dietary pattern (Pattern 3) excludes meat, fish, and rennet-derived cheeses, thus rendering legumes, nuts, milk, yoghurt, certain cheeses, and eggs the primary sources of protein (Figure 6, Table 4). The main part of daily meals is based on vegetables, fruits, and whole grains [407]. This diet is considered healthier than a typical vegetarian or vegan diet, as it provides a small portion of animal protein containing all the necessary amino acids. Similar to the Mediterranean diet, healthy fats can be provided by extra-virgin olive oil and other vegetable oils, as well as by nuts, avocado, or olives. Carbohydrates mainly come from vegetables, fruit, and whole grains, but can also come from honey.

**Figure 6 molecules-30-03666-f006:**
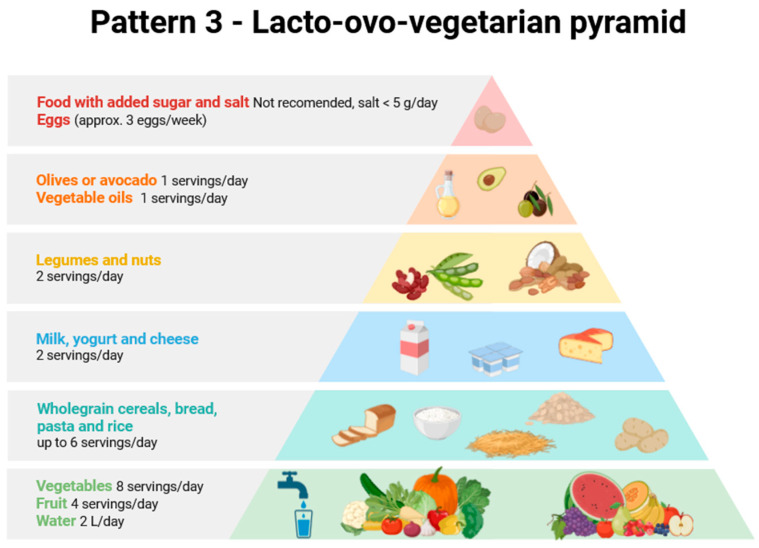
Food pyramid representing a lacto-ovo-vegetarian dietary pattern. Graphic prepared in BioRender [119].

**Table 4 molecules-30-03666-t004:** The lacto-ovo-vegetarian dietary pattern (Pattern 3) adapted to 2000 kcal [407,411,412,413,414].

No	Food Group	Daily Amount	Estimated Energy (kcal)	Energy Percentage (%)	Justification/Comments
1	Vegetables	400 g	100.0	5.0	Low calorie, high volume; approx. 8 servings (½ cup each) of raw and cooked vegetables per day.
2	Fruit	500 g	225	11.2	Approx. 4 servings per day such as 1 cup of fresh and seasonal, colourful fruits, mainly berries; ¾ cup of fruit juice; or ¼ cup of dried fruits.
3	Cereals and grain products	180 g	594	29.7	Main source of energy. Mainly in the form of whole-grain bread, whole-grain pasta, groats, brown rice, and dry cereal. Six servings per day (approx. 30 g each) are recommended.
4	Fats	44 g	396	19.8	Main source of fat; 2 servings per day of vegetable oils rich in monounsaturated fatty acids (14 g per serving) or avocado (50 g) or olives (100 g).
5	Fish and seafood	-	-	-	-
6	Legumes and nuts	100 g + 30 g	300 + 150	22.5	Main source of protein and fibre, and healthy fats from nuts.
7	Dairy	300 g	200.0	10.0	Source of calcium and protein; 2 servings per day, mainly milk, yoghurt, white and curd cheese, and rennet-free ripened cheeses.
8	Eggs	25 g	35	1.8	Provide a small amount of animal protein (½ an egg per day).
9	Meat	-	-	-	-
10a	Water (tap water)	1000 dm^3^	0	0	No energy, but a carrier of MNPs.
10b	Water (bottled)	1000 dm^3^	0	0	No energy, but a carrier of MNPs.
11	Salt	5 g	0	0	The recommended amount is less than 5 g of salt (2 g of Na) per day.
12	Sugar	0 g	0	0	There is no added sugar in the diet; only carbohydrates that are present naturally in the raw materials.

It is certainly important to note that people who are conscious of healthy eating usually prepare meals themselves. Therefore, the MNPs in their diet will come from the raw materials used to prepare the meals, as well as from the kitchen utensils and accessories used for cooking and serving. By contrast, the vast majority of MNPs in the diets of people who consume ready-made meals and takeaway food will be the result of these particles entering food from packaging (containers, cups) and disposable tableware/cutlery.

Based on the assumption and calculations presented above, the average daily intake of MNPs from food has been estimated for three dietary patterns, each providing 2000 kcal (Table 5). The results obtained are similar to those reported by Hussain et al. [230], who demonstrated that heating food in a microwave oven for three minutes releases 4.22 × 10^6^ microplastic particles and 2.11 × 10^9^ nanoplastic particles from every square centimetre of the plastic container’s surface. Similarly, another study showed that the average person consumes 19.38 × 10^9^ microplastic particles per year through the daily intake of 400 g of fruit and vegetables, equating to approximately 53 × 10^6^ particles per day [220]. By contrast, the number of MNP particles consumed through seafood was estimated at 22 × 10^6^ particles per year (i.e., 60 particles/day). Regardless of whether the estimated values are close to the actual values, it is clear that dietary patterns significantly affect the number of MNPs consumed.

Although the Western diet is the least favourable for human health, the low consumption of fruit, vegetables, fish, and seafood means that, in terms of MNP count, this diet performs best (36.4 × 10^6^ particles/day). Despite the lack of MNP-rich fish and seafood, the lacto-ovo-vegetarian diet provides the highest number of MNPs (69.1 × 10^6^ particles/day). This is mainly due to the increased proportion of plant-based ingredients in daily meals.

Considering the number of MNPs in the diet (37.5 × 10^6^ particles/day) and the impact on health, dietary pattern 1, which is based on a vegetarian diet, is the most beneficial. Increasing consumer awareness of how MNPs enter food will further reduce MNP intake from the human diet. Simply drinking tap water instead of bottled water can reduce MNP intake to below 35 × 10^6^ per day. Thoroughly rinsing vegetables and fruit can reduce this number further [284], as can replacing plastic kitchen utensils with glass, wooden, or metal ones.

Figure 7 illustrates the differences in the proportion of individual food groups in daily MNP intake, depending on the dietary pattern. Several food categories, such as fruit and vegetables, as well as bottled water and beverages, have been shown to play a dominant role in this exposure to MNPs. Reducing their share in the diet in favour of MNP-poor food groups would result in reduced plastic particle intake.

As mentioned earlier, an increasing number of studies are attempting to determine dietary MNP intake [415], but most focus on one product group only, e.g., fruit or water, indicating that the values obtained depend heavily on portion size or the amount of product consumed. Consequently, this review is pioneering in its estimation of the level of MNP intake from food depending on a specific dietary pattern.

It is important to be aware of the presence of MNPs in various food categories and to know how to reduce their numbers, because micro- and nanoparticles of plastics can easily penetrate the digestive tract and enter various tissues, where they can have an adverse effect on the body.

## 7. Impact of MNPs on Human Health

As described in previous chapters, micro- and nanoplastic particles can enter the human body from the environment, mainly through ingestion of food and drink, but also by inhalation or through skin penetration. MNPs then spread throughout the body, reaching different locations depending on their size. Particles of PE, PP, PVC, PA6, PA66, PET, PS, ABS, PMMA, and many others have been detected in various human secretions, body fluids, cells, organs, and tissues [416,417,418]. Locations in which MNPs have been found include the human placenta [419,420,421], faeces (including meconium and infant faeces) [420,422,423,424], and various organs, such as the lungs [168,219,425], kidneys, spleen, and cirrhotic liver [426,427], brain [428], heart [429], large and small intestine [430], as well as skin and hair [431]. PE, PP, and PE-co-PP microparticles have been detected in samples taken from the female reproductive tract tissues (ovarian ectopic cysts, uterine tube tissue samples, and adenomyosis) [432], whereas PS, PVC, PE, and PP have been found in the testis [430]. Fluid tissues and secretions also contain MNPs, as do human amniotic fluid [433], saliva [431], sputum [434], urine [435,436], semen [430,437], breast milk [387,420], blood [438,439], and bronchoalveolar lavage fluids [440]. MNPs have also been found in human thrombi [429,441], human erythrocytes [442], atherosclerotic plaques [429,443], various cardiovascular tissues (including the pericardium, myocardium, as well as epicardial, pericardial, and myocardial adipose tissue, the left atrial appendage, and heart muscle) [429,444], human arteries [445], and the saphenous vein [446]. MNPs have a strong affinity for cholesterol, causing the formation of large hetero-aggregates of cholesterol and MNPs. This relationship has been confirmed by the presence of plastic microparticles (including PS, PE, PMMA, PP, PVC, PET, PC, PU, POM, EVA, ABS, and CPE) in all gallstones collected from sixteen patients after cholecystectomy [447]. Zhao et al. [448] have also confirmed the presence of PS, PVC, and PE microparticles in lung, gastric, colorectal, pancreatic, and cervical tumours, but not in oesophageal tumours. Various MNPs, including polyamide, PET, and PVC, have also been identified in tumour and para-tumour tissues of the human prostate [449].

Numerous studies have demonstrated a link between a variety of diseases and MNP pollution, which is associated with their presence in tissues, cells, or organs. This is because MNPs can foster reactions and processes that may harm the human body and promote the development of various diseases and pathologies. For example, MNPs (primarily PS) have been proven to promote oxidative stress [450,451,452], cell senescence and inflammatory response [453,454], as well as DNA damage [429,455] in various cells, and platelet aggregation [456,457], which can lead to disease and premature death.

In animals, PS microparticles of a very small size have been proven to induce a significant increase in cytosolic Ca^2+^ concentration and the enhanced expression of genes encoding proinflammatory cytokines such as IL-8, IL-1β, and TNF-α [450,458]. Exposure to PS has also been reported to activate various signalling pathways leading to oxidative stress and, finally, to increased levels of fibrotic proteins and collagen in a mouse model [451]. Oxidative stress and reactive oxygen species (ROS) generation induced by MNPs contribute to various pathological conditions and reactions, such as inflammation, mitochondrial damage, mitophagy, autophagy, and cell apoptosis [410,458,459,460,461,462], as well as trigger organ damage and systemic toxicity [463]. Pulmonary toxicity [464], cardiotoxicity [465], neurotoxicity [466], nephrotoxicity [462,467], immunotoxicity [461], reproductive toxicity [42], hepatotoxicity [465], genotoxicity [38,468], and DNA damage [452,455] are the result of exposure to MNPs and lead to the development of various health conditions, disorders, and dysfunctions. Exposure to MNPs can cause, among other things, digestive disorders and gastrointestinal tract dysfunction in various animals and humans. This can manifest as altered microbiota composition and dysbiosis [447,469,470], reduced mucus secretion [469,471,472], gut barrier dysfunction and enhanced gut epithelial permeability [472,473,474,475,476], abnormal structure formation in the mouse gut (e.g., gut villus erosion, decreased crypt numbers and depth, and large infiltration of inflammatory cells) [44,473], development of inflammatory bowel disease (IBD) [470,473,477], colitis [478], and colorectal cancer [448,470,473], as well as non-alcoholic fatty liver disease [479], and the induction of hepatic lipid and bile acid disorders [469,472]. Changes in the microbiota, particularly an increase in the abundance of *Veillonella* and *Alistipes* alongside a decrease in *Faecalibacterium* abundance, may lead to obesity [480]. Studies by Silva et al. [410] have also demonstrated the crucial role of mitochondria in both micro- and nanoplastic-induced toxicity, as well as in the pathogenesis of obesity and type 2 diabetes.

It has been reported that small MNPs may modulate the microbiota–gut–brain axis and thus affect physical and mental health [481] and behaviour [481,482]. However, it is the ability of MNPs to cross the blood–brain barrier (BBB) that underlies their neurotoxicity, which manifests itself in the form of oxidative stress, proinflammatory reactions, autophagy, apoptosis, and the inhibition of acetylcholinesterase in the brain [483,484]. Studies on mice have revealed that PS nanoparticles can cross the BBB and reach the brain within only 2 h after oral administration [483]. This neurotoxicity can lead to abnormal protein folding, mitochondrial dysfunction, loss of neurones, and disorders in neurotransmitter function, ultimately resulting in the initiation and progression of neurodegenerative changes, as well as neurobehavioural and neurodevelopmental abnormalities [457,466]. In an in vitro study, chemically inert PE nanoparticles were found to penetrate the 3D structure of a human embryonic stem cell (hESC) model, leading to reduced expression of the HES5, NOTCH1, NEUROD1, and ASCL1 genes [485]. Diminished activity of these genes in mice resulted in severe nervous system impairment, indicating the neurotoxic effect of NPs. When PS nanoparticles were administered orally to mice, they reached the brain, causing cognitive dysfunction [486], as well as deficits in learning and memory [487]. Prenatal exposure of mice to PE led to disturbances in their social interactions and repetitive behaviours, resulting in the development of traits similar to autism spectrum disorder (ASD) in mice [488]. Negatively charged PS nanoparticles have been found to stimulate amyloidogenesis, thereby acting as an exogenous agent that triggers the pathogenesis of Parkinson’s disease. This conclusion was drawn from results showing that NACore (a surrogate for alpha-synuclein, which is associated with Parkinson’s disease pathogenesis) became elevated due to exposure to the nanoplastic via hydrophobic interactions [489]. Furthermore, due to their hydrophobic surface, PS nanoplastics accelerate the aggregation of β-amyloid peptides, which may contribute to the development of Alzheimer’s disease [490].

Chronic exposure to PS nanoparticles was found to significantly increase levels of reactive oxygen species, plasma glucose, and the accumulation of hepatic triglycerides and cholesterol in mice [491]. PS was found to induce the formation of red blood cell aggregates and markedly elevate their adhesion to endothelial cells [492]. The clogging of blood vessels and increased clot formation caused by MNPs [493,494] can underpin cardiovascular disorders. Liu et al. found that MNPs (primarily PET, followed by PA, PVC, and PE) can accumulate in human arteries, which may be associated with atherosclerosis [445]. Patients with MNPs (PE, PVC) in carotid artery atherosclerotic plaques had an increased incidence of cardiovascular events [429], especially a higher risk of heart attack, stroke, or death from any cause during 34 months of follow-up than patients without MNPs [443]. Comprehensive reviews of cardiovascular diseases and dysfunctions induced by MNPs have already been published [495,496].

Microplastic particles have been found to exert a cytotoxic effect, often involving mitochondrial and lysosomal damage, as well as changes in gene expression, inhibition of the electron transport chain, oxidative stress, protein denaturation, haemolysis, and many other effects [148,439,497]. Exposure to PS has been demonstrated to delay the regeneration of skeletal muscle, probably through ROS overproduction [498]. In turn, PE microplastics have been shown to exert genotoxic and cytotoxic effects on human peripheral blood lymphocytes, causing chromosome instability and an elevated frequency of nucleoplasmic bridges [468]. Studies on *Caenorhabditis elegans* have demonstrated that long-term exposure to nanoparticles can cause reproductive toxicity persisting for several generations, even after the NPs have been removed [499]. This trans-generational decline in reproduction has been associated with germline toxicity and epigenetic regulation, and is probably linked to DNA methylation and histone modification. Co-exposure of PS and di-(2-ethylhexyl) phthalic acid has been shown to induce DNA damage, cell cycle arrest, and necroptosis in mouse ovarian granulosa cells by promoting ROS production and resulting oxidative stress [500]. Considering that MNPs can act as vehicles for various toxic compounds or microorganisms, they can also, in this way, increase the risk of genotoxicity, cytotoxicity, and an immunological response. Genomic instability, together with oxidative stress and inflammation, can result in diseases, particularly cancer [320,345,501].

MNPs may affect fertility. Lu et al. have observed that MNPs induced apoptosis of spermatogenic cells in male mice via the p53 signalling pathway [43]. In some cases, the impact on reproduction and fertility is driven by the MNPs’ influence on the endocrine system. Moreover, MNPs act as vectors, carrying endocrine-disrupting chemicals (EDCs) that easily enter organisms. These EDCs act as agonists or antagonists for various hormonal receptors, thereby promoting endocrine toxicity and interfering with the functioning of the hypothalamic–pituitary axis, including regulation of the activity of the hypothalamus, pituitary gland, thyroid, adrenal glands, testes, and ovaries [502,503,504,505,506].

Bearing all this in mind, it is crucial to have knowledge of the harmful effects of MNPs and how to minimise their presence in food and beverages. Implementing this knowledge in homes and food industry enterprises could help to reduce the prevalence of diseases in the future. However, the estimated decomposition time of plastic products (from approx. 450 to 1000 years) and the rapidly increasing amount of plastic items produced in recent decades clearly indicate that the number of micro- and nanoplastics generated from them into the environment will continue to grow for many years to come. This is true regardless of the measures taken. Even if, from today onwards, no plastic ended up in water or soil and all packaging was replaced with fully biodegradable materials, people should be aware that we would still suffer the long-term consequences of exposure to MNP particles for many decades to come. Specifically, these will be related to endocrine system disruption and the resulting problems with fertility or changes in metabolism. This will result in an increasing incidence of diabetes, obesity, various cancers, and problems with conceiving and maintaining pregnancy, as well as other health issues.

## 8. Recommendations for Food Producers and Consumers

Considering the above-mentioned sources of MNPs and activities that increase the number of particles released into the environment, as well as their subsequent entry into the food chain, food products, and human bodies, the following easy-to-implement recommendations and long-term measures are presented below. They would reduce the release of MNPs and limit human exposure to these harmful particles.
(1)In agriculture and horticulture, especially with regard to soils where food crops are grown, we should strive to move away from using plastic film for mulching and sewage sludge for fertilisation. We should also avoid covering fertilisers with plastic.(2)Vegetables and fruit should be thoroughly rinsed with tap water before consumption.(3)Whenever possible, one should avoid drinking bottled water and beverages (both in glass and plastic containers), replacing them with tap water, as well as coffee and tea brewed directly in a glass or ceramic cup. Using tea bags or coffee capsules and drinking from plastic takeaway cups should be limited to exceptional situations, such as travelling.(4)For cooking, baking, and frying, cookware that is suitable for high temperatures and has an undamaged non-stick coating should be used.(5)Wherever possible, plastic kitchen accessories such as cutting boards, spatulas, beaters, spoons, funnels, strainers, whisks, various bowls, kettles, blenders, and mixers should be replaced with glass, cast iron, metal, or wooden ones.(6)Ready meals and raw ingredients should only be heated or defrosted in containers designed for use at high temperatures, such as in a microwave oven.(7)When storing food at home, glass containers are preferable to plastic boxes and bags.(8)Where possible, it is better to prepare meals yourself than to use takeaways or so-called ‘box diets’, as these generate huge amounts of single-use plastic packaging.(9)Food producers should stop wrapping individual fruits and vegetables in plastic film or foil.(10)Retail chains offering fresh goods such as fruit, vegetables, and bread should reduce self-service in these departments to limit the use of individual packaging. Where this is not possible, they should replace plastic packaging with paper or fabric alternatives.(11)As soon as possible, countries and law-making institutions should introduce a simple system of paid collection of glass and plastic packaging (e.g., vending machines or collection points), with plastic packaging being collected based on the weight of the returned items, without the need to scan each individual item. This will encourage consumers to return plastic bottles to collection points instead of throwing them away, which will increase the recovery of this packaging from the market, facilitate its transport for recycling, and reduce environmental pollution.(12)Every manufacturer, particularly those of food products, should strive to abandon plastic packaging in favour of paper or glass. If these materials do not provide adequate protection for the product, they should replace non-degradable plastic packaging with biodegradable packaging (e.g., made of polyhydroxyalkanoates or polylactic acid) as soon as possible.(13)In everyday life, people should focus on buying good-quality clothes that will last as long as possible and thus reduce the number of clothes they own. It is also recommended that dishwashers and washing machines be used efficiently. Increasing the load capacity of washing machines and dishwashers, eliminating pre-washing, and shortening the rinsing cycle are effective ways to reduce the number of MNPs released into the water.(14)We recommend air-drying clothes instead of using tumble dryers, as these release large amounts of microplastics into the air. This will also help to keep your clothes in good condition for longer and reduce your energy consumption.(15)Parents and grandparents are advised to limit the number of plastic toys their children have, replacing them with healthier wooden toys and activities that promote children’s development, such as physical activity and family games, and play.

## 9. Conclusions

The presence of MNPs in the human body is affected by more than just the environment people live in. What is eaten is also important, as are the methods used to prepare and distribute food, the quality of the raw materials used to produce meals, and the proportions in which particular food groups are consumed throughout the day. Even if assuming that individual food groups representing successive levels of the food pyramid contain identical amounts of MNPs, regardless of the diet followed, the different dietary patterns result in different amounts of MNPs being introduced into the body. When considering the number of MNP particles introduced into the body through food alone (estimated at 36.4 × 10^6^ MNP particles per day), the most beneficial diet resembles the typical Western eating pattern characterised by low consumption of fruit, vegetables, fish, and seafood. However, it is worth noting that this diet is very unhealthy for humans and results in an increased incidence of heart disease, hypertension, cancer, obesity, and type 2 diabetes. Despite the absence of fish and seafood, which are rich in MNPs, the lacto-ovo-vegetarian diet is the least beneficial, as it results in the highest consumption of MNPs (69.1 × 10^6^ particles/day). This is mainly due to the very high proportion of fruit, vegetables, legumes, and nuts in daily meals. Taking into account the number of MNPs consumed with meals and their health impact, the healthiest eating pattern is based on the Mediterranean diet. This results in an intake of 37.5 × 10^6^ particles per day and has a very beneficial effect on health. Furthermore, avoiding packaged foods and returning to glass, wooden, or metal kitchen utensils can significantly reduce the number of MNPs consumed. Once introduced into the digestive tract, micro- and nanoplastic particles can easily spread throughout the body. Through oxidative stress, inflammation, and impaired signalling pathways, they may exert a harmful effect and cause various diseases. Therefore, it is extremely important to educate consumers and raise their awareness of the measures they can take to reduce the number of MNP particles they ingest.

## Figures and Tables

**Figure 2 molecules-30-03666-f002:**
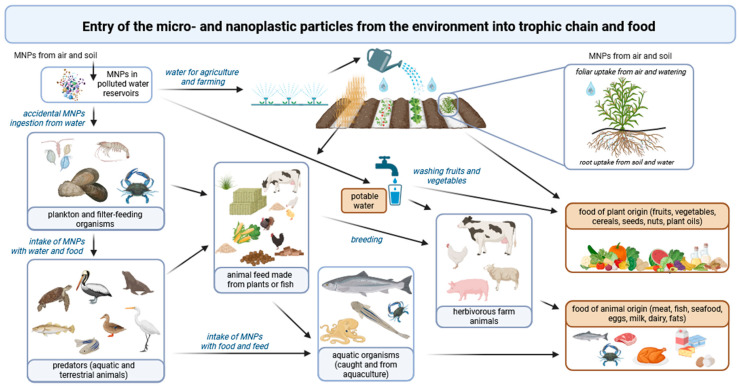
Pathways of the entry of micro- and nanoplastic particles from the environment into the food chain leading to contamination of raw materials used in food production (own compilation, graphic prepared in BioRender [119]).

**Figure 3 molecules-30-03666-f003:**
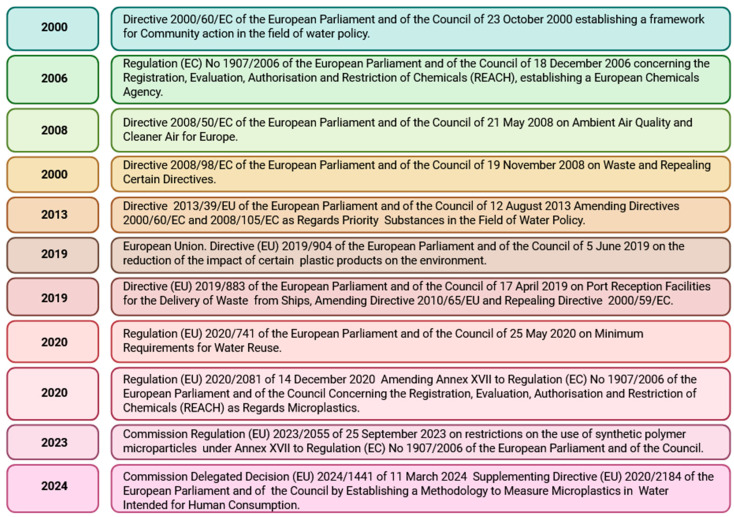
The main legal acts of the European Union aimed at reducing environmental pollution, including MNPs, in chronological order by date of establishment (own compilation, graphic prepared in BioRender [119]).

**Figure 7 molecules-30-03666-f007:**
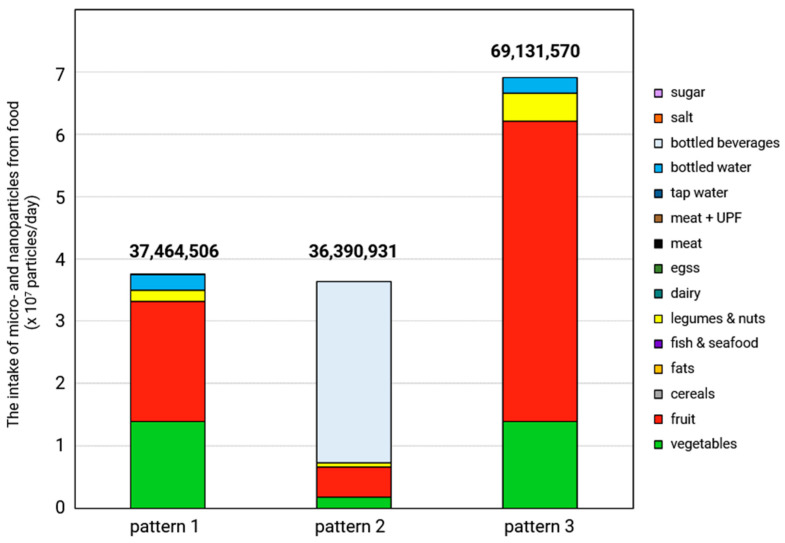
The daily intake of micro- and nanoplastics from food depending on the dietary pattern used. The colours of the bar fragments represent the share of individual food groups in the total MNP intake, depending on the dietary pattern used. Graphic prepared in BioRender [119].

**Table 1 molecules-30-03666-t001:** The reported abundance of micro- and nanoplastic particles present in various food products and categories (where p means MNP particle).

Food Group/Product	Concentration Range of MNPs	Unit	Reference
**Group 1. Vegetables (average value = 34,835 p/g)**			
tomato	3.63 ± 1.39	p/g	[213]
potato	1.5 ± 1.6	p/g	[213]
cucumber	3.6 ± 1.8	p/g	[213]
onion	2.6 ± 1.5	p/g	[213]
lettuce	50,550 ± 25,011	p/g	[380]
lettuce	6.3–29.4	p/g	[283]
carrot	101,950 ± 44,368	p/g	[380]
broccoli	126,150 ± 80,715	p/g	[380]
**Group 2. Fruits (average value = 96,264 p/g)**			
apple	52,600–307,750	p/g	[380]
apple	3.1 ± 1.2	p/g	[213]
pear	98,325–302,250	p/g	[380]
pear	3.1 ± 1.3	p/g	[213]
**Group 3. Cereals (average value = 4.0 p/g)**			
branded flour non-branded flours	2747 ± 654	p/kg	[288]
	6409 ± 625	p/kg	[288]
rice	0.303	p/g	[388]
Indian rice	30.3 ± 8.61	p/100 g	[286]
wheat	4.57	p/g	[388]
**Group 4. Fats and oils (average value = 179 p/cm^3^)**			
edible oil	134,000–580,000	p/dm^3^	[304]
vegetable oils	644–1795	p/dm^3^	[305]
**Group 5. Fish and seafood (average value = 7.14 p/g)**			
mussel	0.206–0.709	p/g	[368]
oysters	0.8–44.1	p/g	[389]
frozen and glazed icefish	0.42 ± 0.28	p/g	[257]
shrimps	0.11	p/g	[293]
shrimps	24 ± 31	p/g	[390]
prawns white leg shrimps	32.66 ± 5.10	p/g	[391]
	10.28 ± 1.19	p/g	[391]
bivalves	0–7.2	p/g	[392]
bivalves	0–10.5	p/g	[393]
crustaceous seafood	0.1–8.6	p/g	[393]
fish	0.0–2.9	p/g	[393]
fish	0.09 ± 0.09	p/g	[394]
fish	0–20	p/g	[392]
canned fish	0.13	p/g	[395]
**Group 6. Legumes and nuts (average value = 34,835 p/g) ***			
**Group 7. Dairy (average value = 4.00 p/g or p/cm^3^)**			
milk	1–14	p/dm^3^	[275]
milk	204–1004	p/100 cm^3^	[274]
milk	1–16	p/dm^3^	[273]
milk	11.1–295.5	p/dm^3^	[388]
conventional butter	942	p/kg	[396]
organic butter	833	p/kg	[396]
sour cream	800	p/kg	[396]
skim milk	134–444	p/dm^3^	[186]
various milks	1–4906	p/100 cm^3^	[387]
yoghurt	20–580	p/dm^3^	[387]
cream	9–596	p/dm^3^	[387]
ayran	18	p/100 cm^3^	[386]
**Group 8. Eggs (average values = 0.12 p/g—assum. mean egg mass = 55 g)**			
eggs	11.67 ± 3.98	p/egg	[303]
**Group 9a. Meat (average value = 0.31 p/g)**			
meat	0.03–1.19	p/g	[228]
packaged meat	4.0–18.7	p/kg	[256]
**Group 9b. Meat + Ultra-processed food (average value = 1.22 p/g)**			
**Takeaway hot beverages (average value = 2.826 p/cm^3^)**			
coffee in			
- PE-coated cups	675–5984	p/dm^3^	[397]
- PP cup	781–4951	p/dm^3^	[397]
- PS cup	838–5215	p/dm^3^	[397]
beverages in single-use plastic cups:			
- PS cup	153−1360	p/dm^3^	[255]
- PE-coated paper cup	126–1346	p/dm^3^	[255]
- EPS cup	246−720	p/dm^3^	[255]
- PP cup	126−1420	p/dm^3^	[255]
beverages in take-out cups:			
- PP cup	1612 ± 216	p/500 cm^3^	[245]
- PET cup	1161 ± 393	p/500 cm^3^	[245]
- PE cup	1482 ± 408	p/500 cm^3^	[245]
- hot drink (60 °C)	1905–2204	p/500 cm^3^	[245]
tea made from teabags	55.6–1446.8	p/250 cm^3^	[398]
**Take-out food (average value = 0.639 p/g)**			
Take-out food	0.639	p/g	[251]
**Group 10a. Water—tap water (average value = 0.22 p/cm^3^)**			
European tap water	0.91–7.73	p/dm^3^	[82]
tap water	0–61	p/dm^3^	[392]
tap water	628	p/dm^3^	[399]
**Group 10b. Water—bottled water (average value = 2527 p/cm^3^)**			
bottled water (all continents)	12.5–2277	p/dm^3^	[264]
water in			
- single-use plastic bottle	2–44	p/dm^3^	[266]
- returnable plastic bottle	28–241	p/dm^3^	[266]
- glass bottle	4–156	p/dm^3^	[266]
mineral water in			
- single-use PET bottle	2648 ± 2857	p/dm^3^	[17]
- reusable PET bottle	4889 ± 5432	p/dm^3^	[17]
- glass bottle	6292 ± 10,521	p/dm^3^	[17]
bottled water	0–6292	p/dm^3^	[392]
bottled water	4889	p/dm^3^	[399]
bottled water	0.07–500	p/dm^3^	[388]
**Group 10c. Bottled non-alcoholic cold beverages (average value = 29,126 p/cm^3^)**			
beverage in cartons	5–20	p/dm^3^	[266]
frozen bottled beverages	68–4.66 × 10^8^	p/dm3	[190]
carbonated bottled beverages:	260.52–281.38	p/dm3	[190]
- with sugar	179–218	p/dm3	[190]
- with additives (sucralose, acesulfame, sunset yellow, and carmine)	157.81–164.24	p/dm3	[190]
soft drink	8.9 ± 2.95	p/dm^3^	[400]
soft drink	0–7 ± 3.21	p/dm^3^	[279]
energy drinks	0–6 ± 1.53	p/dm^3^	[279]
refreshing beverage	68–494	p/dm^3^	[186]
**Group 11. Salt (average value = 0.303 p/g)**			
European sea salt	66.6–220	p/kg	[82]
European sea salt	0–284	p/kg	[392]
lake salt	8–102	p/kg	[392]
rock and well salt	9–185	p/kg	[392]
sea salt	0–1674	p/kg	[401]
lake salt	8–462	p/kg	[401]
rock and well salt	0–204	p/kg	[401]
salt	55.2 ± 43.7 (optically)	p/kg	[216]
	151 ± 61.8 (Nile Red stain)	p/kg	[216]
salt	0.303	p/g	[388]
**Group 12. Sugar (average value = 58.36 p/g)**			
sugar	11,724–53,464 (av. 29,110)	p/100 g	[385]
sugar	57.7 ± 20.6 (optically)	p/kg	[216]
	226 ± 99.5 (Nile Red stain)	p/kg	[216]
sugar	0.343	p/g	[388]
refined sugar	217 ± 123 (fibres)	p/kg	[384]
	32 ± 7 (fragments)	p/kg	[384]

* As no available data on the MNP content in legumes and nuts (expressed as number of MNPs per g) was found, the same value as for vegetables was used for the calculations. The calculations were based on information from studies of rice showing that more MNPs are taken up during the early stages of plant growth (until the grain-setting stage) than during the mature phase (when the rice grains are in the milky phase) [402]. Therefore, we assumed that the amount of MNP in food from group 6. (legumes and nuts) would be lower than in the leaves or roots of vegetables. Given the differences in consumption within this product group, we decided to adopt this value as the maximum possible.

**Table 5 molecules-30-03666-t005:** Estimated average daily intake of micro- and nanoplastic particles from food, depending on the dietary pattern used. The portion sizes of the food groups presented above were taken into account for the different dietary patterns.

No.	Food Group	Average MNPs/g (MNPs/cm^3^)	MNP Intake with Food Depending on Dietary Pattern (MNPs/Food Group)		
			Pattern 1	Pattern 2	Pattern 3
1	Vegetables	34,835	13,934,000	1,741,750	13,934,000
2	Fruit	96,264	19,252,800	4,813,200	48,132,000
3	Cereals	4.00	1000	320	720
4	Fats	179	6265	12,530	7876
5	Fish/seafood	7.14	249.9	45.7	-
6	Legumes/nuts	34,835	1,741,750	696,700	4,528,550
7	Dairy	4.00	1200	160	1200
8	Eggs	0.12	6.6	3	3
9a	Meat	0.31	13.33	-	-
9b	Meat + UPF	1.22	-	317.2	-
10a	Water (tap water)	0.22	220	220	220
10b	Water (bottled)	2527	2,527,000	-	2,527,000
10c	Bottled beverages	29,126	-	29,126,000	-
11	Salt	0.20	1	2	1
12	Sugar	58.36	0	4 698	0
		**total**	**37,464,506**	**36,390,931**	**69,131,570**

## Data Availability

No new data were created or analysed in this study. Data sharing is not applicable to this article.

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
