# Peer review of "The Presence of Micro- and Nanoplastics in Food and the Estimation of the Amount Consumed Depending on Dietary Patterns"

_molecules, 2025, doi:10.3390/molecules30183666_

Round 1

Reviewer 1 Report

Comments and Suggestions for Authors

The review is devoted to a topical issue of our time and has scientific and practical interest. However, there are some comments.

General suggestions

In academic and scientific writing, third-person narration is generally preferred over first-person narration.

In the review, some of the facts and statements presented are not supported by appropriate references.

Comments

Lines 57-60. Numerous studies have indicated the harmful or even toxic effects of MNPs on the morphology, as well   as on the physiological, biochemical and genetic processes of plants, animals and humans, including their reproduction, the functioning of the endocrine system, the immune system and the coexisting microbiota [27]”.

Although the authors mention a significant number of studies on the impact of microplastics on plants, animals, and humans, they provide a reference to only one review, which discusses the effects of microplastics on humans alone.

Lines 61-62. There are no references to support the following statement: “It should also be remembered that various contaminants (organic and inorganic) can enter the body alongside MNPs and negatively influence the living organisms.”

Line 246. The authors cite biodegradation by microorganisms as the primary reason for plastic degradation; however, it is well established that the main reason plastic, microplastic, and nanoplastic accumulate in the environment is their resistance to biodegradation.

Author Response

Our responses to the suggestions and comments of Reviewer #1 are presented in attached file.

Reviewer 2 Report

Comments and Suggestions for Authors

The study molecules-3794422 titled “The presence of micro- and nanoplastics in food products and their amount consumed depending on the type of raw material, the method of food preparation and dietary pattern” by Duda and Petka. Overall, the approach of estimating the ingested quantity of micro- and nanoplastics (MNPs) based on different dietary patterns and food categories is particularly interesting and relevant. However, the manuscript in its current form lacks clarity and coherence. Several sections are redundant or overly detailed, and do not contribute effectively to the main objective of the study. By significantly simplifying the structure and the text, moving the tables to the beginning of the results section, and using them as a basis to describe the findings, the manuscript could become a valuable contribution to research on MNPs as emerging food contaminants. Although the topic is highly relevant and timely, the manuscript requires significant revisions before it can be considered for publication.

Author Response

Our responses to the suggestions and comments of Reviewer #2 are presented in attached file.

Reviewer 3 Report

Comments and Suggestions for Authors

I found the study’s overall design, logical flow, and methodology to be sound upon careful reading. The manuscript would benefit from a thorough review of abbreviation consistency, particularly for commonly referenced terms such as “microplastics,” “micro- and nanoplastics,” and various types of polymers. Below are specific issues noted:

  1. Inconsistent abbreviation of “Micro- and nanoplastics”

The term “Micro- and nanoplastics” should be abbreviated as MNPs after its first use to maintain clarity and conciseness. However, full terms are repeatedly used at:

  • Line 997
  • Line 1085
  • Line 1102
  • Line 1108
  • Line 1163

Please revise to use “MNPs” if the abbreviation has already been defined.

  1. Inconsistent abbreviation of “Microplastics”

The term “Microplastics” should be abbreviated as MPs after its first introduction. However, the full term is used inconsistently in the following lines:

  • Lines: 80, 104, 143, 212, 225, 276, 302, 415, 420, 444, 482, 517, 519, 523, 537, 550, 557, 561, 567, 604, 606, 645, 647, 648, 696, 723, 745, 752, 753, 799, 800

Please revise these instances to use “MPs” if the abbreviation has already been defined in the text.

  1. Inconsistent abbreviation of polymer names

If abbreviations such as PE, PP, and PS are introduced, the full chemical names should not be repeated throughout the manuscript. The following inconsistencies were identified:

  • Polyethylene (PE):
    • Repeated full form in: Lines 279, 285, 298, 384, 494
  • Polypropylene (PP):
    • Repeated full form in: Lines 285, 536, 564, 768
  • Polystyrene (PS):
    • Repeated full form in: Lines 468, 470

  1. Redundant full polymer names after abbreviation already introduced

The following lines include repeated full names of polymers whose abbreviations have already been defined earlier in the text:

  • Lines 668–670: Full forms of PET, HDPE, PVC, LDPE, PP, PS are repeated unnecessarily
  • Line 716: Polysulfone (PSU)
  • Line 739: Polycarbonate (PC)
  • Line 849: Polyoxymethylene (POM)
  • Line 506: Expanded polystyrene (EPS)

Please consider removing redundant full names after abbreviations have been introduced to enhance clarity and avoid confusion.

You can also try drawing pictures in Chapters 2-4.

Author Response

Our responses to the suggestions and comments of Reviewer #3 are presented in attached file.
